# CURRICULUM REINFORCEMENT LEARNING FROM EASY TO HARD TASKS IMPROVES LLM REASONING

**Shubham Parashar**[1][*]  **Shurui Gui**[1][*]  **Xiner Li**[1][*]  **Hongyi Ling**[1]  **Sushil Vemuri**[2]
**Blake Olson**[1]  **Eric Li**[1]  **Yu Zhang**[1]  **James Caverlee**[1]  **Dileep Kalathil**[2]  **Shuiwang Ji**[1][†]
[1]Department of Computer Science & Engineering, Texas A&M University
[2]Department of Electrical & Computer Engineering, Texas A&M University

## ABSTRACT

We aim to improve the reasoning capabilities of language models via reinforcement learning (RL). Recent RL post-trained models like DeepSeek-R1 have demonstrated reasoning abilities on mathematical and coding tasks. However, prior studies suggest that using RL alone to improve reasoning on inherently difficult tasks is less effective. Here, we draw inspiration from curriculum learning and propose to schedule tasks from easy to hard (E2H), allowing LLMs to build reasoning skills gradually. Our method is termed E2H Reasoner. Empirically, we observe that, although easy tasks are important initially, fading them out through appropriate scheduling is essential in preventing overfitting. Theoretically, we establish convergence guarantees for E2H Reasoner within an approximate policy iteration framework. We derive finite-sample complexity bounds and show that, when tasks are appropriately decomposed and conditioned, learning through curriculum stages requires fewer total samples than direct learning. Our code is publicly available at https://github.com/divelab/E2H-Reasoning.

## 1 INTRODUCTION

Large Language Models (LLMs) have demonstrated reasoning capabilities in tasks such as multi-step arithmetic, planning, and code generation. However, the notion of reasoning in LLMs remains vague, with some works equating it to generating intermediate steps during problem solving (Wei et al., 2022; Cobbe et al., 2021). This view, although intuitive, blurs the line between genuine reasoning and surface-level pattern recognition (Stechly et al., 2024; Valmeekam et al., 2023). Therefore, we adopt a view focusing on generalization, defining reasoning *as the ability to extract principles from simpler tasks and apply them to harder ones*. Supporting this capability requires training methods that go beyond imitation and help models learn underlying problem-solving strategies.

In this direction, the success of DeepSeek R1 (Guo et al., 2025) and OpenAI o1 (Jaech et al., 2024) shows that reinforcement learning (RL) based post-training enhances reasoning. RL uses task-specific rewards based on output correctness, unlike supervised fine-tuning (SFT), which trains models to imitate fixed input-output examples (Zelikman et al., 2022). However, RL struggles on harder tasks on which pre-trained models have low zero shot performance (Shao et al., 2024; Zeng et al., 2025) and since rewards are granted only for correct answers, resulting in sparse learning signals.

To address the sparse reward problem, curriculum learning has been applied to reinforcement learning (CRL) by structuring training from easier to harder tasks (Bengio et al., 2009). This idea has recently been extended to LLM post-training (Team et al., 2025; Bercovich et al., 2025). However, these initial efforts primarily rely on simplified strategies, such as training on easy tasks before switching to hard ones after a fixed number of iterations. In contrast, we introduce **E2H Reasoner** (E2H), a CRL approach with a probabilistic scheduler that gradually shifts focus from easy to hard tasks, enabling even LLMs to develop core reasoning abilities and generalize to more complex problems. We show that LLMs can learn tasks they initially failed in the zero-shot setting (Fig. 1). Empirically, E2H achieves state-of-the-art performance across five reasoning tasks, i.e.

---

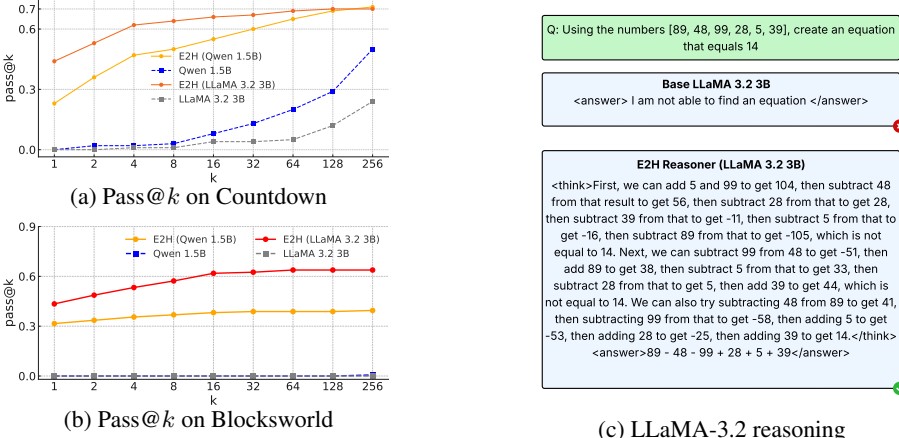

(a) Pass@$k$ on Countdown

(b) Pass@$k$ on Blocksworld

(c) LLaMA-3.2 reasoning

Figure 1: (a, b) Reinforcement learning (RL) based post-training is believed to improve accuracy at low $k$ values in pass@$k$ evaluation (Guo et al., 2025; Yue et al., 2025), we show that **E2H Reasoner**, a curriculum-based RL (CRL) approach, enables LLMs to solve tasks they previously could not, outperforming base models even at higher $k$. (c) LLaMA 3.2 3B reasoning trace for Countdown (Gandhi et al., 2024) after **E2H Reasoner** post-training.

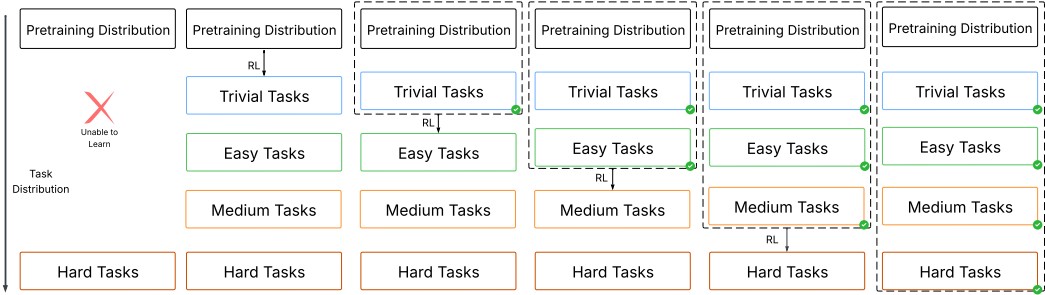

Figure 2: Task Decomposition of **E**asy **2 H**ard Reasoner (E2H). E2H first decomposes the training dataset into tasks of increasing difficulty, namely trivial, easy, and medium, to help the LLM acquire core skills. As training progresses, E2H schedules harder tasks accordingly. See Section 3.2 for scheduling details.

Blocksworld (Valmeekam et al., 2024) and Countdown (Gandhi et al., 2024), as well as three arithmetic reasoning benchmarks (Hendrycks et al., 2021; Agarwal et al., 2021; Cobbe et al., 2021). On the theoretical front, we provide a comprehensive analysis of CRL through the lens of Approximate Policy Iteration, establishing convergence guarantees for the final performance gap and finite-sample complexity bounds. Importantly, we prove that with a well-designed curriculum, CRL requires fewer total samples than direct learning, aligning with and supporting our empirical observations.

## 2 BACKGROUND AND RELATED WORK

**Reasoning in Large Language Models** remains loosely defined, with interpretations varying by tasks and contexts Yu et al. (2024). Prior work describes it as generating logical chains of thought Wei et al. (2022); Wang et al. (2023), performing multi-step deductions (Saparov & He, 2023; Yao et al., 2023; Ling et al., 2025), or simulating human-like problem solving (Parashar et al., 2025). These views, lack clear boundaries between reasoning and pattern recognition. To address this gap, we are inspired by prior work that frames reasoning as generalization or abstraction (Valmeekam et al., 2024; 2023), building on the idea that reasoning involves learning core principles and applying them broadly (Stechly et al., 2024; Huang et al., 2024).

**Post-training of Large Language Models** has emerged as a popular approach to improve reasoning (Snell et al., 2025). These methods are grouped into Supervised Fine-Tuning (SFT) and Reinforcement Learning (RL) based techniques. In SFT, the model is trained to imitate outputs from carefully curated human-like reasoning examples (Zelikman et al., 2022; Muennighoff et al., 2025). However, studies have shown that SFT can lead models to overfit to surface-level patterns (Chu et al., 2025), limiting generalization. In contrast, RL-based post-training uses task-specific rewards and

updates the model through policy optimization algorithms (Schulman et al., 2017; Rafailov et al., 2023; Shao et al., 2024), instead of imitation. This approach has shown greater potential in improving reasoning performance, as demonstrated by the success of models fine-tuned with RL (Guo et al., 2025; Jaech et al., 2024). Still, for inherently difficult tasks that LLMs struggle to solve in zero-shot settings, post-training with RL alone has been insufficient Shao et al. (2024); Zeng et al. (2025).

**Curriculum Learning** organizes tasks by increasing difficulty to promote smoother and more effective training (Bengio et al., 2009; Graves et al., 2017). In the context of RL, it has been applied to help agents acquire complex behaviors by first mastering simpler tasks (Narvekar et al., 2020; Tajwar et al., 2025). Recently efforts have been made to investigate how curriculum-based RL can enhance reasoning and generalization in LLMs (Qiu et al., 2025; Bae et al., 2025; Zeng et al., 2025). Others improve learning by removing examples that are too easy or too hard Bae et al. (2025); Yu et al. (2025), or by maintaining a balanced mix of task difficulties Zeng et al. (2025). Similarly, Chen et al. (2025), and Foster et al. (2025) adaptively sample problems with a $50\%$ solve rate to maximize the GRPO advantage during training. Other recent efforts implement manual curricula switching from easy to hard tasks after a fixed number of training iterations (Xie et al., 2025; Team et al., 2025; Bercovich et al., 2025). In contrast, our work schedules tasks probabilistically from easy to hard, improving reasoning and generalization to out-of-distribution tasks.

## 3 METHOD

**RL for LLM reasoning.** We formulate the reasoning process of LLMs as a RL problem defined over a discounted Markov Decision Process (MDP) $\mathcal{M} = (\mathcal{S}, \mathcal{A}, P, r, \gamma)$, where $\mathcal{S}$ is the state space, $\mathcal{A}$ is the finite action space, $P : \mathcal{S} \times \mathcal{A} \rightarrow \Delta(\mathcal{S})$ is the transition kernel, $r : \mathcal{S} \times \mathcal{A} \rightarrow [0, R_{\max}]$ is the reward function, and $\gamma \in (0, 1)$ is the discount factor. The state space $\mathcal{S}$ consists of all valid token prefixes, where each state $s_t = (x_0, x_1, \ldots, x_t)$ is a sequence of tokens from the vocabulary $\Sigma$. The action space is the vocabulary itself, $\mathcal{A} = \Sigma$. A policy $\pi_\theta$ corresponds to the pre-trained LLM, which defines a distribution over the next token conditioned on the current prefix: $\pi_\theta(x_{t+1}|s_t) = p_\theta(x_{t+1}|x_0, \ldots, x_t)$. We adopt the tags used in (Guo et al., 2025), *i.e.*, <think> </think> and <answer> </answer>, to distinguish intermediate reasoning from final answers. The reward function is sparse: $r(s, a, s') = 0$ for all intermediate states, and $r(s, a, s') = r(y)$ only when the final predicted answer $y$ wrapped by <answer> </answer> is completed. The goal is to optimize the policy to maximize the expected cumulative reward, encouraging the model to generate correct and well-formatted answers through effective reasoning.

### 3.1 TASK DECOMPOSITION FOR RL POST-TRAINING

While SFT provides strong supervision signals, Chu et al. (2025) suggest that reasoning and generalization ability are more effectively enhanced through RL-based post-training. However, applying RL techniques similar to DeepSeek-R1-Zero (Guo et al., 2025) to learn complicated reasoning tasks remains challenging. In this work, we analyze these challenges in two key aspects, the *distribution gap* and *reward design*.

**Challenge 1: Distribution gap.** Learning tasks that exceed the base LLM's reasoning capabilities introduce significant learning challenges. These challenges are often caused by a non-trivial distribution gap between the model pre-training source data distribution $d_0$ and the target data distribution $d_K$. As shown in Fig. 2, because rewards are only given for correct outputs, large distribution shifts can lead to low accuracy and sparse reward signals (Shao et al., 2024; Zeng et al., 2025). Moreover, fitting the model to a single target distribution can lead to overfitting and memorization, undermining the model's generalization and reasoning ability.

**Challenge 2: Reward design.** Challenging reasoning tasks often require LLMs to combine multiple skills to arrive at a solution. While designing a fine-grained, step-by-step reward function could potentially guide the model effectively, such design is generally task-specific and labor-intensive. For example, a computer science student is supposed to learn basic mathematics and linear algebra before learning machine learning. Similarly, a typical Countdown (Gandhi et al., 2024) task involves skills like basic arithmetic, estimating the distance to the goal, and backtracking. While it is possible to include a supervision signal for each intermediate step, doing so is not sustainable and generalizable across diverse tasks.

To overcome these RL learning challenges, we propose task decomposition by splitting training data into subsets of increasing difficulty, based on either human annotations or model performance. This aligns with curriculum learning, where we interpolate between the pre-training distribution $d_0$ and target task distribution $d_K$ via intermediate stages $d_k{}_{k=1}^{K}$, reducing distribution shift and improving training stability (see Fig. 2). From the reward design perspective, decomposing tasks by difficulty breaks complex skill acquisition into simpler steps. For instance, the 6-number Countdown task involves using six integers and four operators $(+, -, \times, \div)$ to reach a target number, requiring the model to perform arithmetic, estimate distances to the goal, and backtrack effectively. In contrast, a 2-number problem focuses primarily on arithmetic, allowing the model to build foundational competence before scaling to harder variants. This avoids handcrafted reward shaping and improves transferability.

For tasks like Blocksworld, MATH, and Countdown, we use human-aligned difficulty signals such as plan length, labeled difficulty, or operand count (see Table 9). For others like AQuA and GSM8K, where such annotations are unavailable, we automatically estimate difficulty based on model error rates under CoT prompting and group examples accordingly (see Figs. 5 and 6). We generate 20 responses per question using 1-shot CoT prompting, with prompts and in-context examples from (Hao et al., 2024; Parashar et al., 2025), and assign difficulty by grouping questions based on quartiles of the error rate distribution.

In this work, we tackle the challenge of learning a hard reasoning task by partitioning the training distribution into four difficulty levels, namely, trivial, easy, medium, and hard, then we adapt our pre-trained LLM to these tasks in a sequential curriculum. In Section 3.3, we provide a theoretical justification that, under fixed sampling and training resources, learning step by step leads to better performance than training directly on the hard task.

### 3.2 Training schedulers for LLM reasoning ability

While task decomposition simplifies RL post-training, traditional curriculum learning poses two main challenges, mainly task forgetting and overfitting, due to the rigid progression through tasks in a fixed task order.

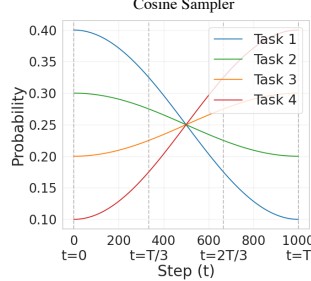

Figure 3: Illustration of cosine scheduling.

**Challenges: Task forgetting and overfitting.** The first challenge, task forgetting, refers to the degradation in performance on earlier (easier) tasks as the model adapts to later (harder) tasks. According to the traditional model generalization literature (Arjovsky et al., 2019; Gulrajani & Lopez-Paz, 2021), the task distribution shifts are considered as explicit signals for the model generalization direction; thus, retaining strong performance across all task distributions is essential for generalization. Therefore, the task forgetting will undermine the model's generalization capability, *i.e.*, the reasoning ability. Task overfitting, the second challenge, arises when the model overfits to trivial tasks and prefers simplistic patterns or short answers that bypass meaningful reasoning. This phenomenon is called reward hacking (Laidlaw et al., 2025), where the model exploits shortcuts on easy tasks and fails to learn harder ones, resulting in poor reasoning performance. To address both challenges, we explore different training sampling techniques, forming four different scheduling strategies as follows.

**Traditional scheduling.** We first formulate the traditional sequential curriculum learning sampler with $T$ training steps as $\mathbf{S}_{\text{trad}}(t, k) = 1$ when $\tau_k \leq t \leq \tau_{k+1}$, otherwise, 0, where $t$ denotes the current training step; for $K$ tasks $k = 1, \ldots, K$, $\tau_k$ denotes the threshold when the curriculum learning proceeds to the $k$-th stage, while $\tau_1 = 0$ and $\tau_{K+1} = T$. The output of the sampler denotes the probability of sampling data from the $k$-th task; therefore, at the $t$ step, the sampling distribution will be $[\mathbf{S}_{\text{trad}}(t, 1), \mathbf{S}_{\text{trad}}(t, 2), \ldots, \mathbf{S}_{\text{trad}}(t, K)]$.

**Balanced scheduling.** To avoid forgetting, the simplest way is to mix all data with different difficulties together and sample randomly, which can be considered as a trivial case of curriculum learning. Alternatively, this can be interpreted as the default behavior of any policy optimization algorithm (Shao et al., 2024; Schulman et al., 2017), where training occurs without considering task difficulty. This balanced sampler can be written as: $\mathbf{S}_{\text{balanced}}(t, k) = \frac{1}{K}$, where each task difficulties have the same probability to be selected at each training step. Although this is an efficient way to avoid forgetting, this sampling introduces harder tasks too early, leading to sparser rewards and suboptimal CRL.

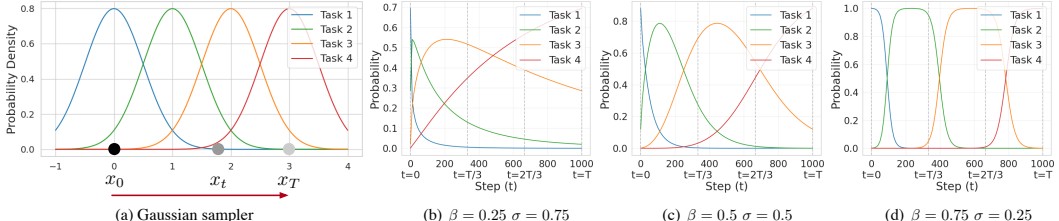

Figure 4: **Gaussian Sampler.** (a) This figure represents the Gaussian sampling process. (bcd) These figures denote the sampling probabilities of different tasks changing along the training steps with different Gaussian sampler hyperparameters.

**Cosine scheduling (E2H-C).** To alleviate both the reward sparsity and forgetting issues, we propose a non-parametric scheduling strategy, namely, cosine sampling. This strategy can be written as: $\mathbf{S}_{\text{cosine}}(t,k) = \alpha_t \cdot (K-k-1) + (1-\alpha_t) \cdot k$, and $\alpha_t = 0.5 \cdot (1 + \cos(\frac{\pi t}{T}))$, where the resulting probabilities need to be normalized before sampling. Intuitively, this cosine sampler sets both the initial and ending sampling probabilities simply according to their ordinal number and interpolates the intermediate probabilities using a cosine function. As shown in Figure 3, the easiest task has the highest probability of being sampled at the beginning, and has the lowest probability of being sampled at the end.

**Gaussian scheduling (E2H-G).** Although cosine scheduling addresses reward sparsity and forgetting, the parameter-free design limits flexibility in handling issues like trivial task overfitting and fine-grained control over different learning stages. Empirically, while adding the trivial task can boost the model performance, it is also easy for the model to overfit to trivial tasks. To overcome this challenge, we propose a Gaussian scheduling strategy inspired by the Gaussian mixture model (Reynolds et al., 2009).

As shown in Figure 4 (a), in a one-dimensional space, we assume the data distributions of tasks follow Gaussian distributions with the same variance $\sigma$. The means of the adjacent task's Gaussian distributions are assumed to have the same distance 1, *i.e.*, $\mu_k = k-1$. Then the sampling probability is defined as the likelihood of a given position $x_t$ belonging to different task Gaussian distributions, similar to the expectation–maximization algorithm (McLachlan & Krishnan, 2008). Therefore, the Gaussian scheduling strategy can be expressed as:

$$\mathbf{S}_{\text{Gaussian}}(t,k) = \exp\left(-\frac{(x_t - \mu_k)^2}{2\sigma^2}\right), \text{ and } x_t = \left(\frac{t}{T}\right)^{\beta}(K-1), \tag{1}$$

where we ignore the normalization term, and the probabilities will be normalized for sampling. In this sampling scheduler, we only have two hyperparameters, *i.e.*, $\sigma$ and $\beta$. While the variance $\sigma$ controls the sampling concentration, $\beta > 0$ controls the sampling position $x_t$'s moving speed. When $\sigma$ is smaller, the training is more focused, more similar to traditional curriculum learning. When $\beta < 1$, the sampling process will assign fewer training steps focusing on easier tasks and train harder tasks longer, avoiding easy task overfitting. As shown in Figure 4 (bcd), we use three typical hyperparameter settings for this Gaussian sampler, *i.e.*, $\beta = 0.25\ \sigma = 0.75$, $\beta = 0.75\ \sigma = 0.25$, and $\beta = 0.5\ \sigma = 0.5$.

### 3.3 THEORETICAL ANALYSIS

In this section, we analyze our curriculum reinforcement learning framework for LLMs under the lens of Approximate Policy Iteration (API). We follow the theoretical structure of Chen & Maguluri (2025); Scherrer (2014), adapting it to our curriculum setting. Specifically, we show how sequentially solving interpolated curriculum distributions enables convergence guarantees and improved sample complexity bounds, under function approximation errors. Our analysis considers action-value functions and explicitly tracks the impact of approximation and distribution shift across curriculum stages. All proofs are presented in Appendix A.

#### 3.3.1 THEORETICAL SETUP

Recall our MDP $\mathcal{M} = (\mathcal{S}, \mathcal{A}, P, r, \gamma)$. We denote the policy space as $\Pi$, where each policy $\pi \in \Pi$ is a mapping from states to distributions over actions, $\pi : \mathcal{S} \to \Delta(\mathcal{A})$. In our curriculum setting, we introduce a sequence of MDPs $\{\mathcal{M}_k\}_{k=1}^{K}$ where each MDP shares the same state and action spaces

but may differ in reward functions and/or transition dynamics. Each MDP induces a state visitation distribution $d_k$ under the optimal policy for that curriculum, with $\{d_k\}_{k=1}^K$ interpolating between an easy source distribution $d_1$ and the final hard task $d_K$. Let $\pi_k$ be the learned policy at curriculum step $k$, and let $\pi_K^*$ be the optimal policy under the final target task. The goal is to learn a sequence of policies $\{\pi_k\}_{k=1}^K$ such that the final policy $\pi_K$ performs well under the target distribution $d_K$.

The action-value function of a policy $\pi$ is defined as $Q^\pi(s, a) := \mathbb{E}\left[\sum_{t=0}^\infty \gamma^t r(s_t, a_t) | s_0 = s, a_0 = a, a_t \sim \pi(\cdot|s_t), s_{t+1} \sim P(\cdot|s_t, a_t)\right]$. Let $\mathcal{T}^\pi$ be the Bellman operator for policy $\pi$, defined as $\mathcal{T}^\pi Q(s, a) := r(s, a) + \gamma \mathbb{E}_{s' \sim P(\cdot|s,a), a' \sim \pi(\cdot|s')}\left[Q(s', a')\right]$. We define the optimal Bellman operator $\mathcal{T}$, $\mathcal{T}Q(s, a) := r(s, a) + \gamma \mathbb{E}_{s' \sim P(\cdot|s,a)}\left[\max_{a'} Q(s', a')\right]$. The fixed point of the optimal Bellman operator is $\mathcal{T}Q^* = Q^*$.

The Approximate Policy Iteration (API) framework is a generalization of classical policy iteration that accommodates function approximation and inexact updates. It serves as a foundational tool for analyzing practical reinforcement learning algorithms including actor-critic and deep RL methods. The API algorithm alternates between two stages. At iteration $k$, the algorithm performs two algorithmic steps. $(i)$ *Policy Evaluation:* Given policy $\pi_k$, compute an approximate estimate $\widehat{Q}_k$ of its action-value function $Q^{\pi_k}$, $(ii)$ *Policy Improvement:* Update the policy to $\pi_{k+1}$, which is greedy or approximately greedy with respect to $\widehat{Q}_k$. In the context of CRL, API offers a natural framework to study how sequentially adapted policies, defined over interpolated curriculum distributions, evolve toward the optimal policy and shape the final policy $\pi_K$.

We adopt the following API assumptions from Chen & Maguluri (2025); Scherrer (2014), adapted to the curriculum setting. Let $Q_k := Q^{\pi_k}$ be the action-value function for the policy $\pi_k$ at curriculum $k$.

**Approximate Policy Evaluation.** At each iteration $k$, a function approximator $\widehat{Q}_k$ is used to estimate $Q_k$. The estimated Q-function $\widehat{Q}_k^{\pi_k}$ satisfies $\|\widehat{Q}_k^{\pi_k} - Q_k^{\pi_k}\|_\infty \leq \delta_k$.

**Approximate Greedy Policy Improvement.** Let $\pi_{k+1}$ be an $\epsilon_k$-greedy policy with respect to $\widehat{Q}_k$: $\mathbb{E}_{s \sim \mu_k}\left[Q^{\pi_{k+1}}(s, \pi_{k+1}(s))\right] \geq \mathbb{E}_{s \sim \mu_k}\left[\max_a \widehat{Q}_k(s, a)\right] - \epsilon_k$, for a sampling distribution $\mu_k$ at step $k$, *i.e.*, $\|\mathcal{T}\widehat{Q}_k^{\pi_k} - Q_k^{\pi_{k+1}}\|_\infty \leq \epsilon_k$.

**Distribution Mismatch (Concentrability).** Let $\mu_k$ be the sampling distribution for policy improvement at step $k$, and $d_k$ the true task distribution. We assume the distribution mismatch between $\mu_k$ and $d_k$ is bounded by $C_k$, *i.e.*, $\sup_{s \in \mathcal{S}} \frac{d_k(s)}{\mu_k(s)} \leq C_k$.

**Bounded Curriculum Drift.** For the weighted norm, $\|Q\|_{d_k} = \sqrt{\mathbb{E}_{s \sim d_k}[\max_a Q(s, a)^2]}$. The deviation between successive optimal Q-functions satisfies $\|Q_K^* - Q_k^*\|_{d_K}$ is bounded for all $k$.

### 3.3.2 CONVERGENCE GUARANTEE

Let $Q_K^*$ be the optimal Q-function for the final task under distribution $d_K$, and let $Q^{\pi_K}$ be the Q-function of the policy learned at the final step. Define the performance loss of the final policy compared to the optimal target policy as: $\mathcal{E}_K := \|Q_K^* - Q^{\pi_K}\|_{d_K}$.

**Theorem 3.1** (CRL Performance Guarantee). *Let $T$ be the number of API policy updates within each task. $\beta > 0$ is a tunable parameter for stepsizes specified in Chen & Maguluri (2025). Under the approximate greedy update and evaluation error assumptions above, the final performance gap $\mathcal{E}_K$ satisfies:*

$$\mathcal{E}_K \leq \sum_{k=1}^K \left(\gamma^T \eta_k + \frac{2\gamma(1 - \gamma^T)}{(1 - \gamma)^2}\delta_k + \frac{2\gamma}{\beta(1 - \gamma)^2}\right) + \sum_{k=1}^{K-1} \|Q_K^* - Q_k^*\|_{d_K},$$

*where $\eta_k := \|Q_k^* - Q_k^{\pi_k}\|_\infty$ is the per-task Bellman error, $\delta_k$ is the evaluation error, and $\epsilon_k$ is absorbed in $\eta_k$.*

The first term represents the convergence bias of the actor, and goes to zero geometrically fast as $T$ goes to infinity. The second term captures accumulated evaluation errors, which involves the stochastic error due to sampling and the error due to function approximation The third term captures the error introduced by the policy update rule, which can be made arbitrarily small by using large enough $\beta$. Alternatively, we can use geometrically increasing stepsizes, in which case the third term goes to zero at a geometric rate. The last term captures the deviation in optimal Q-functions

across curriculum stages, representing the cumulative gap between intermediate curriculum-optimal values and the final optimal value, which we refer to as the curriculum approximation error. This decomposition highlights the dual effect of CRL in improving sample efficiency (small $\delta_k, \epsilon_k$) and ensuring smooth interpolation (small $\|Q_K^* - Q_k^*\|$).

### 3.3.3 FINITE SAMPLE APPROXIMATION ERROR ANALYSES

In this section, we perform finite-sample analysis. Considering the critic, i.e., how to obtain an estimate of $Q^\pi$, we can estimate the Q-function $Q^\pi$ of a given target policy $\pi$ using TD-learning. In TD-learning, especially when the state-action space size is large, the use of function approximation is natural. In linear function approximation, a set of basis vectors is selected to approximate the target value function $Q^\pi$ using linear combinations of the basis vectors. Let $\Phi \in \mathbb{R}^{|\mathcal{S}||\mathcal{A}| \times d}$ be the matrix of basis vectors $\Phi = [\phi_1, \cdots, \phi_d]$. Then, the goal is to find from the linear subspace $\mathcal{Q} = \{\hat{Q}_w = \Phi w \mid w \in \mathbb{R}^d\}$ the "best" approximation of the Q-function $Q^\pi$, where $w \in \mathbb{R}^d$ is the weight vector.

Following Section 3 and 4 of Chen & Maguluri (2025), we derive the finite-sample theorem for CRL. Theorem A.1 and its analyses are provided in Appendix A.1.1. The result of Theorem A.1 suggests that smoother curriculum trajectories, together with smaller approximation errors under finite samples, can lead to improved final policy performance. These effects are closely tied to the choice of stepsize and the number of updates per iteration.

Following Chen & Maguluri (2025), let $\mathcal{K}_{\mathcal{SA}} \in \mathbb{R}^{|\mathcal{S}||\mathcal{A}| \times |\mathcal{S}||\mathcal{A}|}$ be a diagonal matrix and let $\mathcal{K}_{\mathcal{SA},\min}$ be the minimal diagonal entry. $J_k$ is the number of critic updates per policy iteration at curriculum $k$. The bootstrapping parameter $n$ is chosen such that $\gamma_c := \gamma^n / \sqrt{\mathcal{K}_{\mathcal{SA},\min}} < 1$. $L_k$ is the parameter defined by $1 + (\gamma \rho_{\max,k})^n$, $\mathcal{E}_{\text{approx},k} := \sup_\pi \left\| Q_{c,\rho}^{\pi_k} - \Phi w_{c,\rho}^{\pi_k} \right\|_\infty$ is the critic's function approximation error, $\|Q_K^* - Q_k^*\|_\infty$ is the gap between the curriculum subtask and the final task, and $\lambda_{\min}$ is the mininum eigenvalue of the positive definite matrix $\Phi_T \mathcal{K}_{\mathcal{SA}} \Phi$. The term $N_{2,1} = \sum_{k=1}^K \frac{2\gamma \mathcal{E}_{\text{approx},k}}{(1-\gamma)^2}$ in Theorem A.1 is the function approximation error and equals zero when using a complete basis. A term of similar form is present in all existing work studying RL with function approximation (Munos, 2003; Agarwal et al., 2021). Based on Theorem A.1, we next derive the sample complexity result.

**Theorem 3.2** (Sample Complexity). *For a given accuracy level $\epsilon > 0$, to achieve $E[\|Q_K^* - Q^{\pi_K}\|_\infty] \leq \epsilon + N_{2,1} + \sum_{k=1}^{K-1} \|Q_K^* - Q_k^*\|_{d_K}$, the total number of samples (i.e., the integer $T \sum_{k=1}^K J_k$) required across all $K$ curriculum stages is of the order:*

$$O\left( \sum_{k=1}^K \frac{\log^3(1/\epsilon_k)}{\epsilon_k^2} \cdot \tilde{O}\left( \frac{L_k^2 n}{(1-\gamma)^7 (1-\gamma_c)^3 \lambda_{min}^3} \right) \right),$$

*where $\epsilon_k$ is the target accuracy for curriculum $k$, and $\sum_{k=1}^K \epsilon_k \leq \epsilon$.*

*Let $M_{CRL} = \sum_{k=1}^K M_k$ be the total number of samples needed by CRL with $K$ curriculum steps to achieve error $\epsilon$ on the final task, and let $M_{Direct} = m * M_K$ be the number of samples needed by direct learning, where $m > 1$ represents the relative difficulty factor of direct learning. Under geometric error and $L_k$ allocation $\epsilon_k = \epsilon_K \cdot e^{K-k}$ and $L_k = \frac{L_K}{l^{K-k}}$ for curriculums, we have:*

$$M_{CRL} < M_{Direct} \iff \frac{(e*l)^{2(1-K)} - 1}{1 - (e*l)^2} < m - 1. \tag{2}$$

The geometric error and $L_k$ allocation reflect the curriculum gradually increases in difficulty while allowing larger errors in earlier stages. Since the function $f(x) = \frac{x^{1-K} - 1}{1 - x^2}$ monotonically decrease for $x > 1$ when $K$ is a integer larger than 1, the final condition can be reasonably satisfied, *e.g.*, with $K = 3$, $e * l = 1.4$, and $m = 1.8$ in practice.

The first half of Theorem 3.2 analysis highlights the dual benefit of curriculum design in CRL. First, by constructing intermediate distributions $d_1, \ldots, d_{K-1}$ close to $d_K$, the curriculum error term $\|Q_K^* - Q_k^*\|$ can be made small. Second, easier curriculum tasks improve estimation accuracy and yield more stable approximate greedy updates, enhancing sample efficiency and policy improvement. For the second half of Theorem 3.2, the final condition in Eq. 2 is satisfied when curriculums are appropriately learned, the allocation of accuracy targets $\epsilon_k$ is gradually optimized across curriculum

Table 1: Impact of task decomposition for LLM post-training. Trivial and easy examples help the model learn core principles that enable success on harder tasks.

| | Blocksworld | | | | | Countdown | | | | | MATH | | | | |
|---|---|---|---|---|---|---|---|---|---|---|---|---|---|---|---|
| | Trivial | Easy | Med | Hard | OOD | Trivial | Easy | Med | Hard | OOD | Trivial | Easy | Med | Hard | OOD |
| Hard | 0.0 | 0.0 | 0.0 | 0.0 | 0.0 | 0.0 | 43.9 | 16.4 | 18.1 | 6.5 | 82.3 | 64.7 | 53.4 | 38.2 | 20.6 |
| Med + Hard | 2.0 | 0.0 | 0.0 | 0.0 | 0.0 | 12.9 | 47.8 | 33.1 | 19.2 | 8.8 | 87.1 | 68.9 | 58.1 | 42.5 | 21.0 |
| Easy + Med + Hard | 0.0 | 55.5 | 15.5 | 0.0 | 0.0 | 62.5 | 79.3 | 30.1 | 21.2 | 9.5 | 84.8 | 68.2 | 56.8 | 42.5 | 21.5 |
| Trivial + Easy + Med + Hard | 98.0 | 100 | 83.3 | 21.1 | 2.6 | 96.1 | 64.9 | 28.8 | 18.1 | 9.2 | 87.2 | 72.0 | 61.6 | 46.3 | 25.7 |

Table 2: Effect of scheduling strategies in LLM post-training. We compare balanced scheduling, traditional curriculum learning (CL), and our proposed **E2H Reasoner** variants, namely, E2H-G and E2H-C. CoT is reported as a reference.

| | Blocksworld | | | | | Countdown | | | | | MATH | | | | |
|---|---|---|---|---|---|---|---|---|---|---|---|---|---|---|---|
| | Trivial | Easy | Med | Hard | OOD | Trivial | Easy | Med | Hard | OOD | Trivial | Easy | Med | Hard | OOD |
| CoT | 4.0 | 0.0 | 0.0 | 0.0 | 0.0 | 16.0 | 5.6 | 1.7 | 0.1 | 0.1 | 40.1 | 27.9 | 22.7 | 17.6 | 8.2 |
| Balanced | 98.0 | 100 | 84.5 | 26.3 | 5.3 | 96.1 | 64.9 | 28.8 | 18.1 | 9.2 | 87.2 | 72 | 61.6 | 46.3 | 25.7 |
| CL | 46.0 | 100 | 45.2 | 5.8 | 0.7 | 57.7 | 85.8 | 57.2 | 31.5 | 12.6 | 86.2 | 71.5 | 62.4 | 46.7 | 25.6 |
| E2H-G (0.25, 0.75) | 98.0 | 100.0 | 95.3 | 32.9 | 7.3 | 98.9 | 87.3 | 51.4 | 18.9 | 7.3 | 85.5 | 72 | 64.1 | 47.9 | 26.5 |
| E2H-G (0.5, 0.5) | 100 | 100 | 34.5 | 10.5 | 0.7 | 97.9 | 87.2 | 70.4 | 41.0 | 19.2 | 85.3 | 71.7 | 62.5 | 48.7 | 27.6 |
| E2H-G (0.75, 0.25) | 98.0 | 93.3 | 17.9 | 2.0 | 0.0 | 95.7 | 56.0 | 28.8 | 17.1 | 10.2 | 86.0 | 72.4 | 62.0 | 46.7 | 26.3 |
| E2H-C | 100 | 100 | 15.5 | 0.0 | 0.0 | 96.7 | 64.0 | 25.9 | 15.8 | 6.4 | 84.6 | 69.6 | 63.0 | 47.6 | 28.6 |

steps, and the designed curriculums with increasing difficulties effectively bridge the gap between source and target distributions. This mathematical derivation shows CRL requires fewer total samples than direct learning on the final task, aligning with experimental observations (see Appendix B).

# 4 EXPERIMENTS

We conduct experiments investigating the following research questions. **RQ1:** What role does task decomposition play in RL-based post-training? **RQ2:** How does task scheduling impact the learning process? **RQ3:** Can small-scale LLMs learn to reason on hard tasks?

## 4.1 EXPERIMENTAL SETUP

We evaluate our method on a diverse set of reasoning and planning tasks, covering both datasets with and without human-annotated difficulty levels. For datasets with difficulty labels, such as Blocksworld (using plan length)(Valmeekam et al., 2023), Countdown (using number of operations)(Gandhi et al., 2025), and MATH (using problem levels) (Hendrycks et al., 2021), we use the provided annotations and define an out-of-distribution (OOD) split to assess generalization. For datasets without explicit difficulty, namely GSM8K (Cobbe et al., 2021) and AQuA (Ling et al., 2017a), we estimate difficulty using the zero shot error rate of the base model. Specifically, for each question in the training set, we generate 20 responses and compute the error rate as $1 - \frac{\text{Number of Correct Responses}}{20}$ and bucket the samples into trivial, easy, medium, and hard, based on quartiles Fig. 5. More dataset details are in Appendix C. We conduct experiments on Qwen 2.5/1.5B Instruct (Yang et al., 2024), LLaMa 3.2 3B Instruct (Grattafiori et al., 2024). More models are included in Appendix G.1. We use Qwen 1.5B for research questions RQ1 and RQ2.

## 4.2 BASELINES

First, we report the Chain of Thought (Wei et al., 2022) (**CoT**) performance for all models. All post-training experiments use GRPO (Shao et al., 2024) as the reinforcement learning algorithm (see Appendix E for implementation). GRPO, by default, employs balanced scheduling over all tasks, which we use as our baseline and refer to as **GRPO** in Table 3. We also assess whether models can learn directly from the most challenging examples by training only on the **Hard** and **OOD** subsets. In addition, we evaluate traditional curriculum learning (**CL**) as used by Team et al. (2025); Bercovich et al. (2025). We compare against **Self-Evolve** (Chen et al., 2025), an adaptive curriculum baseline that samples problems with a $50\%$ solve rate to maximize learnability. Finally, we provide a comparison against SFT in Appendix F.

## 4.3 EXPERIMENTAL RESULTS

We conduct experiments addressing the research questions listed above. We examine how task decomposition impacts LLM post-training using Qwen-1.5B-Instruct (RQ1) with a balanced scheduler (Table 1). We find that including trivial and easy examples helps the model build core skills. This

Table 3: Results of **E2H Reasoner** across three models on Blocksworld (Valmeekam et al., 2023), Countdown (Gandhi et al., 2024) and MATH Hendrycks et al. (2021). Our method consistently improves performance especially on HARD and OOD tasks, demonstrating effective reasoning, results on more models are in Appendix G.1. Best numbers are in **bold** and second-best are underlined.

| | | Blocksworld | | | | | Countdown | | | | | MATH | | | | |
|---|---|---|---|---|---|---|---|---|---|---|---|---|---|---|---|---|
| | | Trivial | Easy | Med | Hard | OOD | Trivial | Easy | Med | Hard | OOD | Trivial | Easy | Med | Hard | OOD |
| | CoT | 4.0 | 0.0 | 0.0 | 0.0 | 0.0 | 16.0 | 5.6 | 1.7 | 0.1 | 0.1 | 40.1 | 27.9 | 22.7 | 17.6 | 8.2 |
| | GRPO (All) | 98.0 | **100** | 83.3 | 21.1 | 2.6 | 96.1 | 64.9 | 28.8 | 18.1 | 9.2 | **87.2** | **72.0** | 61.6 | 46.3 | 25.7 |
| | GRPO (Hard) | 0.0 | 0.0 | 0.0 | 0.0 | 0.0 | 0.0 | 43.9 | 16.4 | 18.1 | 6.5 | 82.3 | 64.7 | 53.4 | 38.2 | 20.6 |
| Qwen 1.5B Instruct | GRPO (OOD) | 0.0 | 0.0 | 0.0 | 0.0 | 0.0 | 3.1 | 23.1 | 18.1 | 11.3 | 5.3 | 37.0 | 21.1 | 15.0 | 8.5 | 3.7 |
| | CL | 46.0 | **100** | 45.2 | 5.8 | 0.7 | 57.7 | 85.8 | 57.2 | 31.5 | 12.6 | 86.2 | 71.5 | 62.4 | 46.7 | 25.6 |
| | Self Evolve | **100** | **100** | 70.2 | 13.8 | 2.1 | 96.6 | 65.3 | 34.2 | 17.8 | 9.5 | 84.0 | 70.6 | 62.6 | 48.6 | 26.1 |
| | E2H-G | 98.0 | **100** | **95.3** | **32.9** | **7.3** | **97.9** | **87.2** | **70.4** | **41.0** | **19.2** | 85.3 | 71.7 | 62.5 | **48.7** | 27.6 |
| | E2H-C | **100** | **100** | 15.5 | 0.0 | 0.0 | 96.7 | 64.0 | 25.9 | 15.8 | 6.4 | 84.6 | 69.6 | **63.0** | 47.6 | **28.6** |
| | CoT | 24.0 | 0.0 | 1.2 | 1.0 | 0.0 | 37.1 | 4.6 | 0.3 | 0.0 | 0.0 | 65.9 | 44.6 | 35.2 | 24.1 | 13.6 |
| | GRPO (All) | **100** | **100** | 94.1 | 38.9 | 13.3 | 99.9 | 89.5 | 71.6 | **47.9** | 2.7 | 65.9 | 47.0 | 36.0 | 22.0 | 10.2 |
| | GRPO (Hard) | 0.0 | 0.0 | 0.0 | 0.0 | 0.0 | 40.5 | 33.8 | 3.0 | 9.7 | 1.4 | 22.7 | 14.4 | 10.3 | 7.5 | 0.3 |
| LLaMa 3.2 3B Instruct | GRPO (OOD) | 0.0 | 0.0 | 0.0 | 0.0 | 0.0 | 8.0 | 0.0 | 0.0 | 0.0 | 0.0 | 63.6 | 39.0 | 31.3 | 19.0 | 7.6 |
| | CL | **100** | 0.0 | 0.0 | 0.0 | 0.0 | 17.2 | 36.0 | 22.7 | 11.2 | 4.1 | 74.1 | 54.1 | 43.9 | 28.0 | 12.5 |
| | Self-Evolve | **100** | **100** | 91.1 | 35.8 | 16.6 | 96.7 | 66.6 | 37.9 | 27.5 | 18.5 | **79.1** | **61.4** | **48.9** | 33.1 | 14.1 |
| | E2H-G | **100** | **100** | **98.8** | **44.1** | **17.6** | 95.0 | **89.9** | **73.3** | 46.5 | **24.3** | 78.7 | 58.4 | 46.4 | 32.3 | 14.5 |
| | E2H-C | **100** | 0.0 | 0.0 | 0.0 | 0.0 | 100 | 55.3 | 0.0 | 0.0 | 0.0 | 74.8 | 60.6 | 48.3 | **34.3** | **15.8** |

Table 4: Performance of **E2H Reasoner** on GSM8K and AQuA, where difficulty splits are derived from error rates due to the absence of human labels. Fig. 5 shows these splits, and Table 15 confirms robustness to the number of splits..

Figure 5: GSM8K difficulty distribution based on error rates. Difficulty groups used for training are derived from quartiles of this distribution.

| **Qwen 1.5B Instruct** | | | | | | | | | | |
|---|---|---|---|---|---|---|---|---|---|---|
| | **GSM8K** | | | | | **AQuA** | | | | |
| | Trivial | Easy | Med | Hard | Avg | Trivial | Easy | Med | Hard | Avg |
| CoT | 90.2 | 87.3 | 76.5 | 38.1 | 67.7 | 70.8 | 51.3 | 20.8 | 2.6 | 40.9 |
| GRPO | **99.0** | 95.3 | 84.1 | 49.9 | 77.1 | **95.8** | 68.0 | 48.6 | 21.0 | 63.3 |
| CL | 98.0 | **97.2** | 85.8 | **52.2** | 78.6 | 88.8 | 72.2 | 36.1 | 18.4 | 58.6 |
| Self-Evolve | 98.1 | 95.3 | 87.0 | 50.3 | 77.8 | 94.4 | 75.0 | 40.3 | 31.6 | 64.2 |
| E2H-G | 97.6 | 94.7 | **89.0** | 51.8 | **78.7** | 90.2 | **81.9** | 43.0 | **34.2** | **66.1** |
| E2H-C | 98.0 | 95.3 | 83.9 | 46.6 | 75.7 | 86.1 | 72.2 | **48.6** | 26.3 | 62.5 |

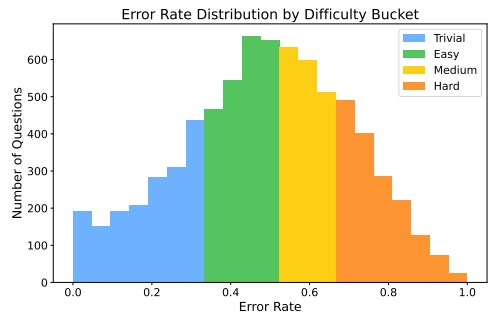

enables effective transfer from simpler to harder tasks and better OOD performance, consistent with our view of reasoning as learning core principles and applying them to harder tasks.

Next, we examine how scheduling impacts post-training (RQ2) in Table. 2. Balanced scheduling serves as a strong baseline but lacks structure, leading to suboptimal learning. CL schedules tasks in a fixed difficulty order, which can cause forgetting of earlier tasks or overfitting to easier ones. In our experiments, easier tasks help initiate learning due to their dense rewards, but overexposure hinders generalization to harder, sparse-reward tasks. We address this through our two schedulers, i.e., E2H-C and E2H-G (see Section 3.2). For tasks like MATH, where models perform reasonably well across all difficulty levels (as seen with CoT in Table 2), the cosine scheduling in E2H-C is effective, beginning with a focus on trivial and easy tasks and gradually shifting toward harder ones. However, on tasks like Blocksworld, where rewards are sparse since harder examples are more challenging, this leads to overfitting and degraded performance. E2H-G addresses this issue by using a Gaussian schedule that quickly decays the sampling probability of trivial and easy tasks. As shown in Fig. 4, it provides enough exposure early on to support initial learning while rapidly shifting focus to harder examples. This prevents overfitting and improves generalization in sparse-reward settings where E2H-C struggles.

Next, we investigate whether models can learn to reason directly from difficult examples (RQ3). As shown in Table 3, the answer is largely no. For instance, Qwen-2.5 1.5B when trained on Level 5 MATH directly, underperforms the CoT baseline on the model! This failure highlights the need for CRL methods for LLM reasoning. To this end, we compare with a CRL method, Self-Evolve (Table 3). Empirically, we find that its reasoning performance varies significantly across models and tasks. Self-Evolve during training, samples problems at a 50% success rate to maximize learnability and gradually reduces easier ones once this threshold is surpassed. However, being only halfway proficient at solving a problem is often not enough. For example, learning calculus requires strong

Table 5: Ablation of E2H on GRPO vs DAPO. E2H improves overall performance over both baselines and yields consistent gains when combined with DAPO, indicating that the two approaches are complementary.

| Method | Variant | Blocksworld | | | | | Countdown | | | | | MATH | | | | |
|---|---|---|---|---|---|---|---|---|---|---|---|---|---|---|---|---|
| | | Triv. | Easy | Med. | Hard | OOD | Triv. | Easy | Med. | Hard | OOD | Triv. | Easy | Med. | Hard | OOD |
| GRPO | Baseline | 98 | 100 | 83.3 | 21.1 | 2.6 | 96.1 | 64.9 | 28.8 | 18.1 | 9.2 | 87.2 | 72.0 | 61.6 | 46.3 | 25.7 |
| | E2H-G | 98 | 100 | 95.3 | 32.9 | 7.3 | 97.9 | 87.2 | 70.4 | 41.0 | 19.2 | 85.3 | 71.7 | 62.5 | 48.7 | 27.6 |
| | E2H-C | 100 | 100 | 15.5 | 0.0 | 0.0 | 96.7 | 64.0 | 25.9 | 15.8 | 6.4 | 84.6 | 69.6 | 63.0 | 47.6 | 28.6 |
| DAPO | Baseline | 100 | 100 | 89.3 | 24.3 | 3.4 | 96.5 | 67.3 | 35.7 | 22.0 | 10.9 | 85.6 | 73.1 | 63.3 | 47.9 | 28.3 |
| | E2H-G | 100 | 100 | 98.8 | 46.7 | 9.3 | 99.9 | 90.6 | 82.4 | 59.4 | 30.1 | 86.3 | 72.7 | 64.4 | 49.1 | 28.5 |
| | E2H-C | 100 | 100 | 96.4 | 44.1 | 8.7 | 96.5 | 64.1 | 26.0 | 16.7 | 8.7 | 86.3 | 72.3 | 64.5 | 50.2 | 29.2 |

Table 6: Ablation of E2H on GRPO vs DAPO on GSM8K and AQuA.

| Method | Variant | AQUA | | | | | GSM8k | | | | |
|---|---|---|---|---|---|---|---|---|---|---|---|
| | | Triv. | Easy | Med. | Hard | Avg | Triv. | Easy | Med. | Hard | Avg |
| GRPO | Baseline | 95.8 | 68.0 | 48.6 | 21.0 | 63.3 | 99.0 | 95.0 | 84.1 | 49.9 | 77.1 |
| | E2H-G | 90.2 | 81.9 | 43.0 | 34.2 | 66.1 | 97.6 | 94.7 | 89.0 | 51.8 | 78.7 |
| | E2H-C | 86.1 | 72.2 | 48.6 | 26.3 | 62.5 | 98.0 | 95.3 | 83.9 | 46.6 | 75.7 |
| DAPO | Baseline | 94.4 | 72.2 | 48.6 | 31.5 | 65.7 | 97.0 | 98.0 | 87.5 | 52.0 | 79.0 |
| | E2H-G | 95.8 | 81.9 | 51.3 | 28.9 | 69.3 | 99.1 | 97.3 | 89.3 | 53.2 | 80.1 |
| | E2H-C | 95.8 | 75.0 | 55.5 | 36.8 | 69.6 | 98.5 | 97.0 | 88.1 | 48.8 | 78.0 |

Table 7: Pass@k Performance on AIME24 and OlympiadBench for models trained on MATH.

| Model | Method | AIME24 | | OlympiadBench | |
|---|---|---|---|---|---|
| | | Pass@1 | Pass@32 | Pass@1 | Pass@32 |
| Qwen 1.5B | GRPO | 3.3 | 10.0 | 7.3 | 33.3 |
| | Self Evolve | 3.3 | 10.0 | 8.0 | 37.0 |
| | E2H-G | 6.7 | 16.7 | 8.7 | 39.3 |
| | E2H-C | 6.7 | 16.7 | 10.0 | 40.0 |
| LLaMA 3B | GRPO | 0.0 | 3.3 | 3.7 | 22.0 |
| | Self Evolve | 3.3 | 10.0 | 5.3 | 29.3 |
| | E2H-G | 3.3 | 10.0 | 5.3 | 28.6 |
| | E2H-C | 6.7 | 10.0 | 6.0 | 30.0 |

command of school mathematics. In addition, implementing Self-Evolve also requires careful tuning of hyper-parameters for different models and datasets.

In contrast, **E2H Reasoner** guides learning scheduling tasks from easy to hard, improving generalization, as reflected in stronger OOD performance (RQ3). This is further supported by the results, where **E2H Reasoner** shows better performance over baselines as task difficulty rises. Note that for E2H-G we report the best numbers out of our 3 parameter settings with extensive results in Table 13 and does not need extensive hyper-parameter tuning. Finally, **E2H Reasoner** remains effective even without human difficulty labels by using error rates as a proxy for difficulty (Table 4).

In Table 5, we compare against DAPO, which builds on GRPO by filtering out problems that are either too easy (all rewards = 0) or too hard (all rewards = 1). In principle, if DAPO were allowed to resample up to $N$ times (where $N$ is the dataset size) before forming each batch, it would always find the perfect batch, but this is computationally infeasible for training. Our **E2H Reasoner** mitigates this issue by gradually focusing on harder task groups through probabilistic difficulty-based scheduling. As shown in Figure 7, combining E2H with DAPO reduces the the fraction of batches with zero advantage during training, indicating that it selects prompts whose overall difficulty is better aligned with the model's current competence. Taken together, these results show that E2H and DAPO are complementary; E2H shapes the difficulty conditioned sampling distribution that DAPO resamples from, and their combination yields the strongest performance across benchmarks.

Finally, we conduct an extended evaluation of our models trained on the MATH dataset using E2H and vanilla GRPO on two challenging evaluation benchmarks, AIME24 (Veeraboina, 2023) and OlympiadBench (He et al., 2024). E2H demonstrates stronger generalization to both tasks in Table 7

## 5 CONCLUSION

We introduce the **E2H Reasoner** (E2H), a CRL-based method for LLM post-training. E2H enables models to learn tasks they initially failed at by scheduling tasks from easy to hard. E2H challenges the assumption that small LLMs cannot reason and demonstrates strong empirical performance supported by theoretical analysis, offering convergence guarantees and improved sample efficiency over direct RL. In summary, E2H provides a scalable, theoretically grounded, and practical method for LLM reasoning.

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

## APPENDIX

## A PROOFS

We now present the proofs for the main theoretical results stated above. Our analysis closely follows the derivation style of Chen & Maguluri (2025); Scherrer (2014), extending it to the curriculum setting.

### A.1 PROOF OF THEOREM 3.1

*Proof.* We aim to derive a tight bound for the final performance error in task $\mathcal{T}_K$:

$$\mathcal{E}_K := \|Q_K^* - Q_K^{\pi_K}\|_{d_K},$$

where $Q_K^*$ is the optimal Q-function for the final task $\mathcal{T}_K$, and $Q_K^{\pi_K}$ is the Q-function under the learned policy $\pi_K$.

We begin by analyzing the policy performance error under API. For each task $k$, we define:

$$\eta_k := \|Q_k^* - Q_k^{\pi_k}\|_\infty.$$

By norm monotonicity, we have:

$$\|Q_K^* - Q_K^{\pi_K}\|_{d_K} \leq \|Q_K^* - Q_K^{\pi_K}\|_\infty.$$

We proceed by recalling the result from Theorem 2.1 of Chen & Maguluri (2025) for a fixed task $k$, applied with a step size $\beta$:

$$\|Q_k^* - Q_k^{\pi_k}\|_\infty \leq \gamma^T \|Q_k^* - Q_k^{\pi_k^0}\|_\infty + \frac{2\gamma}{1-\gamma} \sum_{t=0}^{T-1} \gamma^{T-1-t} \delta_k + \frac{2\gamma}{\beta(1-\gamma)^2},$$

where $\delta_k$ is the approximate value evaluation error at curriculum step $k$; $\gamma^T \|Q_k^* - Q_k^{\pi_k^0}\|_\infty$ reflects initialization at $\pi_k^0$; and $\beta > 0$ is the API soft update step size parameter. The finite geometric series

$$\sum_{t=0}^{T-1} \gamma^{T-1-t}$$

has closed-form expression

$$\sum_{i=0}^{T-1} \gamma^i = \frac{1-\gamma^T}{1-\gamma} \text{ for } \gamma \neq 1 \quad \text{and} \quad \sum_{i=0}^{T-1} \gamma^i = 1, \text{ for } \gamma = 1.$$

Thus we simplify with

$$\sum_{i=0}^{T-1} \gamma^i = \frac{1-\gamma^T}{1-\gamma}.$$

Since we denote:

$$\eta_k := \|Q_k^* - Q_k^{\pi_k}\|_\infty;$$

then, the bound becomes:

$$\eta_k \leq \gamma^T \eta_k + \frac{2\gamma(1-\gamma^T)}{(1-\gamma)^2}\delta_k + \frac{2\gamma}{\beta(1-\gamma)^2}.$$

We isolate $\eta_k$:

$$\eta_k(1-\gamma^T) \leq \frac{2\gamma(1-\gamma^T)}{(1-\gamma)^2}\delta_k + \frac{2\gamma}{\beta(1-\gamma)^2} \Rightarrow \eta_k \leq \frac{1}{1-\gamma^T}\left(\frac{2\gamma(1-\gamma^T)}{(1-\gamma)^2}\delta_k + \frac{2\gamma}{\beta(1-\gamma)^2}\right).$$

We want to bound:

$$\|Q_K^* - Q_K^{\pi_K}\|_{d_K};$$

using triangle inequality:

$$\|Q_K^* - Q_K^{\pi_K}\|_{d_K} \leq \|Q_K^* - Q_K^{\pi_K}\|_\infty \leq \eta_K + \sum_{k=1}^{K-1}\|Q_K^* - Q_k^*\|_{d_K}.$$

Let us now plug in the bound for $\eta_K$:

$$\|Q_K^* - Q_K^{\pi_K}\|_{d_K} \leq \frac{1}{1-\gamma^T}\left(\frac{2\gamma(1-\gamma^T)}{(1-\gamma)^2}\delta_k + \frac{2\gamma}{\beta(1-\gamma)^2}\right) + \sum_{k=1}^{K-1}\|Q_K^* - Q_k^*\|_{d_K}.$$

The distribution mismatch is defined by $C_k := \left\|\frac{d_K}{d_k}\right\|_\infty$. If $d_k$ is constructed by curriculum to smoothly interpolate toward $d_K$, and both $d_K$ and $d_k$ are supported on the same or growing state space. In our analysis, we use:

$$\|Q_k^* - Q_k^{\pi_k}\|_{d_K} \leq C_k \cdot \|Q_k^* - Q_k^{\pi_k}\|_\infty,$$

where $\|\cdot\|_{d_K}$ is the $L_2$ norm under the distribution $d_K$ and $\|\cdot\|_\infty$ is the sup norm (worst-case).

In general, for any probability distribution $\mu$, and any function $f$,

$$\|f\|_{L_2(\mu)} = \left(\int f(x)^2 d\mu(x)\right)^{1/2} \leq \|f\|_\infty.$$

So, in fact, we have the reverse:

$$\|f\|_{d_K} \leq \|f\|_\infty \quad \Rightarrow \quad C_k \leq 1.$$

If we do want to write

$$\|f\|_{d_K} \leq C_k \cdot \|f\|_\infty,$$

then the tightest possible value for $C_k$ is exactly 1, and any value $C_k < 1$ may be possible depending on the support of $f$ under $d_K$. This holds when later curriculum stages subsume earlier ones, a design principle in CRL. Hence, using $C_k \leq 1$ simplifies bounds without loosening them unnecessarily.

Because curriculum proceeds sequentially, and assuming that earlier tasks are easier and learned more accurately, we can sum the per-step bounds for curriculum steps:

$$\mathcal{E}_K = \|Q_K^* - Q_K^{\pi_K}\|_{d_K} \leq \sum_{k=1}^{K}\left[\frac{1}{1-\gamma^T}\left(\frac{2\gamma(1-\gamma^T)}{(1-\gamma)^2}\delta_k + \frac{2\gamma}{\beta(1-\gamma)^2}\right)\right] + \sum_{k=1}^{K-1}\|Q_K^* - Q_k^*\|_{d_K}.$$

This completes the proof.

Optionally, assuming $\gamma^T$ is small for large $T$, $1/(1-\gamma^T) \leq 2$ leads to:

$$\mathcal{E}_K \lesssim \sum_{k=1}^{K}\left(\frac{4\gamma(1-\gamma^T)}{(1-\gamma)^2}\delta_k + \frac{4\gamma}{\beta(1-\gamma)^2}\right) + \sum_{k=1}^{K-1}\|Q_K^* - Q_k^*\|_{d_K}.$$

$\square$

### A.1.1 THEOREM A.1 AND PROOF

**Theorem A.1** (Finite-Sample Guarantee). *Consider the sequence of tasks $\{\mathcal{T}_k\}_{k=1}^K$, where the final task $\mathcal{T}_K$ has optimal action-value function $Q_K^*$. Suppose that $\pi_K$ is the final policy produced by CRL and assumptions hold for all curriculum stages. Following Chen & Maguluri (2025), let $\mathcal{K}_{\mathcal{SA}} \in \mathbb{R}^{|\mathcal{S}||\mathcal{A}| \times |\mathcal{S}||\mathcal{A}|}$ be a diagonal matrix and let $\mathcal{K}_{\mathcal{SA},\min}$ be the minimal diagonal entry. $J_k$ is the number of critic updates per policy iteration at curriculum $k$. The bootstrapping parameter $n$ is chosen such that $\gamma_c := \gamma^n / \sqrt{\mathcal{K}_{\mathcal{SA},\min}} < 1$. Then, when using constant stepsize $\alpha$ satisfying $\alpha(t_\alpha + n + 1) \leq \frac{(1-\gamma_c)\lambda_{\min}}{130 L^2}$, we have for all $J_k \geq t_\alpha + n + 1$:*

$$
\mathbb{E}\left[\|Q_K^* - Q^{\pi_K}\|_\infty\right] \leq \underbrace{\gamma^{TK}\|Q_K^* - Q^{\pi_0}\|_\infty}_{N_1} + \sum_{k=1}^K \left(\underbrace{\frac{2\gamma \mathcal{E}_{approx,k}}{(1-\gamma)^2}}_{N_{2,1}} + \underbrace{\frac{2\gamma^2 \mathcal{E}_{bias,k}}{(1-\gamma)^4}}_{N_{2,2}}\right)
$$

$$
+ \underbrace{\sum_{k=1}^K \frac{6(1-(1-\gamma_c)\lambda_{min}\alpha)^{\frac{1}{2}[J_k-(t_\alpha+n+1)]}}{(1-\gamma)^3(1-\gamma_c)^{1/2}\lambda_{min}^{1/2}}}_{N_{2,3}} + \underbrace{\sum_{k=1}^K \frac{70 L_k[\alpha(t_\alpha+n+1)]^{1/2}}{\lambda_{\min}(1-\gamma_c)(1-\gamma)^3}}_{N_{2,4}}
$$

$$
+ \underbrace{\frac{2\gamma\beta K}{(1-\gamma)^2}}_{N_3} + \underbrace{\sum_{k=1}^{K-1} \|Q_K^* - Q_k^*\|_{d_K}}_{\text{Curriculum discrepancy}},
$$

*where $c(\cdot,\cdot)$ and $\rho(\cdot,\cdot)$ are generalized importance sampling factors, $L_k$ is the parameter defined by $1 + (\gamma\rho_{max,k})^n$, $\mathcal{E}_{approx,k} := \sup_\pi \left\|Q_{c,\rho}^{\pi_k} - \Phi w_{c,\rho}^{\pi_k}\right\|_\infty$ is the critic's function approximation error, $\mathcal{E}_{bias,k} := \max_{0 \leq t \leq T} \max_{s \in \mathcal{S}} (1-\lambda(s)) \|\pi_{k,t}(\cdot|s) - \pi_{k,b}(\cdot|s)\|_1$ is the importance sampling bias at curriculum $k$, $\|Q_K^* - Q_k^*\|_\infty$ is the gap between the curriculum subtask and the final task, and $\lambda_{\min}$ is the mininum eigenvalue of the positive definite matrix $\Phi_T \mathcal{K}_{\mathcal{SA}}\Phi$.*

The terms $N_1$ and $N_3$ are the same as appeared in Thm. 3.1, and together capture the error in actor update. Term $N_{2,1}$ is the function approximation error and equals zero when using a complete basis. A term of similar form is present in all existing work studying RL with function approximation (Munos, 2003; Agarwal et al., 2021). Term $N_{2,2}$ is the bias introduced by generalized importance sampling factors $c(\cdot,\cdot)$ and $\rho(\cdot,\cdot)$, and $N_{2,2} = 0$ when $c(s,a) = \rho(s,a) = \pi(a|s)/\pi_b(a|s)$. Term $N_{2,3}$ represents the convergence bias in the critic and goes to zero geometrically fast as the inner loop iteration index $J_k$ goes to infinity. The term $N_{2,4}$ represents the variance in the critic and is proportional to $\sqrt{\alpha t_\alpha} = \mathcal{O}(\sqrt{\alpha \log(1/\alpha)})$, thus arbitrarily small given small enough stepsize $\alpha$. Together, $\{N_{2,i}\}_{i=1\sim 4}$ correspond to the second term in Thm. 3.1. Finally, the curriculum approximation error is the same as the last term in in Theorem 3.1. This result suggests that smoother curriculum trajectories and smaller approximation errors under finite samples, which is closely related with the stepsize and number of updates per iteration, can lead to better final policy performance.

*Proof.* We proceed by mathematical induction across the curriculum steps. Let $\pi_{k,T}$ denote the final policy after $T$ iterations at curriculum step $k$. **Base case:** For the first curriculum step ($k = 1$), we directly apply Theorem 4.1 from Chen & Maguluri (2025):

$$
\mathbb{E}[\|Q_1 - Q^{\pi_{1,T}}\|_\infty] \leq \gamma^T \|Q_1 - Q_1^{\pi_0}\|_\infty \tag{3}
$$

$$
+ \frac{2\gamma E_{approx,1}}{(1-\gamma)^2} + \frac{2\gamma^2 E_{bias,1}}{(1-\gamma)^4} \tag{4}
$$

$$
+ \frac{6(1-(1-\gamma_c)\lambda_{min}\alpha)^{\frac{1}{2}[J_1-(t_\alpha+n+1)]}}{(1-\gamma)^3(1-\gamma_c)^{1/2}\lambda_{min}^{1/2}} \tag{5}
$$

$$
+ \frac{70 L_1[\alpha(t_\alpha+n+1)]^{1/2}}{\lambda_{min}(1-\gamma_c)(1-\gamma)^3} + \frac{2\gamma\beta}{(1-\gamma)^2} \tag{6}
$$

**Inductive step:** Assume that for curriculum step $k$, we have:

$$\mathbb{E}[\|Q_k - Q^{\pi_k,T}\|_\infty] \leq \gamma^{Tk}\|Q_1 - Q_1^{\pi_0}\|_\infty$$
$$+ \sum_{j=1}^{k} \left( \frac{2\gamma E_{approx,j}}{(1-\gamma)^2} + \frac{2\gamma^2 E_{bias,j}}{(1-\gamma)^4} \right.$$
$$+ \frac{6(1-(1-\gamma_c)\lambda_{min}\alpha)^{\frac{1}{2}[J_j-(t_\alpha+n+1)]}}{(1-\gamma)^3(1-\gamma_c)^{1/2}\lambda_{min}^{1/2}}$$
$$\left. + \frac{70 L_j[\alpha(t_\alpha+n+1)]^{1/2}}{\lambda_{min}(1-\gamma_c)(1-\gamma)^3} + \frac{2\gamma\beta}{(1-\gamma)^2} \right)$$
$$+ \sum_{j=1}^{k-1} \|Q_k - Q_j\|_{d_k}$$

For curriculum step $k+1$, we initialize with policy $\pi_{k,T}$ and apply Theorem 4.1 from Chen & Maguluri (2025):

$$\mathbb{E}[\|Q_{k+1} - Q^{\pi_{k+1},T}\|_\infty] \leq \gamma^T\|Q_{k+1} - Q_{k+1}^{\pi_k,T}\|_\infty$$
$$+ \frac{2\gamma E_{approx,k+1}}{(1-\gamma)^2} + \frac{2\gamma^2 E_{bias,k+1}}{(1-\gamma)^4}$$
$$+ \frac{6(1-(1-\gamma_c)\lambda_{min}\alpha)^{\frac{1}{2}[J_{k+1}-(t_\alpha+n+1)]}}{(1-\gamma)^3(1-\gamma_c)^{1/2}\lambda_{min}^{1/2}}$$
$$+ \frac{70 L_{k+1}[\alpha(t_\alpha+n+1)]^{1/2}}{\lambda_{min}(1-\gamma_c)(1-\gamma)^3} + \frac{2\gamma\beta}{(1-\gamma)^2}$$

We need to relate $\|Q_{k+1}^* - Q_{k+1}^{\pi_k,T}\|_\infty$ to our induction hypothesis. By triangle inequality:

$$\|Q_{k+1} - Q_{k+1}^{\pi_k,T}\|_\infty \leq \|Q_{k+1} - Q_k\|_\infty + \|Q_k - Q_k^{\pi_k,T}\|_\infty + \|Q_k^{\pi_k,T} - Q_{k+1}^{\pi_k,T}\|_\infty$$
$$\leq \|Q_{k+1} - Q_k\|_\infty + \|Q_k^* - Q_k^{\pi_k,T}\|_\infty + \frac{\gamma}{1-\gamma}\|r_k - r_{k+1}\|_\infty$$

where the last inequality follows from the performance difference lemma with respect to rewards. For curriculum learning, we design the reward functions to satisfy $\|r_k - r_{k+1}\|_\infty \leq \delta_r$ for some small $\delta_r$. Thus:

$$\|Q_{k+1} - Q_{k+1}^{\pi_k,T}\|_\infty \leq \|Q_{k+1} - Q_k\|_\infty + \|Q_k - Q_k^{\pi_k,T}\|_\infty + \frac{\gamma\delta_r}{1-\gamma}$$

Substituting our induction hypothesis:

$$
\mathbb{E}[\|Q_{k+1} - Q^{\pi_{k+1},T}\|_\infty] \leq \gamma^T \mathbb{E}[\|Q_{k+1} - Q_k\|_\infty + \|Q_k - Q_k^{\pi_k,T}\|_\infty + \frac{\gamma \delta_r}{1-\gamma}]
$$
$$
+ \frac{2\gamma E_{approx,k+1}}{(1-\gamma)^2} + \frac{2\gamma^2 E_{bias,k+1}}{(1-\gamma)^4}
$$
$$
+ \frac{6(1-(1-\gamma_c)\lambda_{min}\alpha)^{\frac{1}{2}[J_{k+1}-(t_\alpha+n+1)]}}{(1-\gamma)^3(1-\gamma_c)^{1/2}\lambda_{min}^{1/2}}
$$
$$
+ \frac{70 L_{k+1}[\alpha(t_\alpha+n+1)]^{1/2}}{\lambda_{min}(1-\gamma_c)(1-\gamma)^3} + \frac{2\gamma}{\beta(1-\gamma)^2}
$$
$$
\leq \gamma^T \|Q_{k+1} - Q_k\|_\infty + \gamma^T \mathbb{E}[\|Q_k^* - Q_k^{\pi_k,T}\|_\infty] + \frac{\gamma^{T+1}\delta_r}{1-\gamma}
$$
$$
+ \frac{2\gamma E_{approx,k+1}}{(1-\gamma)^2} + \frac{2\gamma^2 E_{bias,k+1}}{(1-\gamma)^4}
$$
$$
+ \frac{6(1-(1-\gamma_c)\lambda_{min}\alpha)^{\frac{1}{2}[J_{k+1}-(t_\alpha+n+1)]}}{(1-\gamma)^3(1-\gamma_c)^{1/2}\lambda_{min}^{1/2}}
$$
$$
+ \frac{70 L_{k+1}[\alpha(t_\alpha+n+1)]^{1/2}}{\lambda_{min}(1-\gamma_c)(1-\gamma)^3} + \frac{2\gamma\beta}{(1-\gamma)^2}
$$

Applying the induction hypothesis:

$$
\mathbb{E}[\|Q_{k+1} - Q^{\pi_{k+1},T}\|_\infty] \leq \gamma^T \|Q_{k+1} - Q_k\|_\infty + \gamma^T \cdot \gamma^{Tk} \|Q_1 - Q_1^{\pi_0}\|_\infty
$$
$$
+ \gamma^T \sum_{j=1}^{k} \left( \frac{2\gamma E_{approx,j}}{(1-\gamma)^2} + \frac{2\gamma^2 E_{bias,j}}{(1-\gamma)^4} \right.
$$
$$
+ \frac{6(1-(1-\gamma_c)\lambda_{min}\alpha)^{\frac{1}{2}[J_j-(t_\alpha+n+1)]}}{(1-\gamma)^3(1-\gamma_c)^{1/2}\lambda_{min}^{1/2}}
$$
$$
\left. + \frac{70 L_j[\alpha(t_\alpha+n+1)]^{1/2}}{\lambda_{min}(1-\gamma_c)(1-\gamma)^3} + \frac{2\gamma}{\beta(1-\gamma)^2} \right)
$$
$$
+ \gamma^T \sum_{j=1}^{k-1} \|Q_k - Q_j\|_{d_k} + \frac{\gamma^{T+1}\delta_r}{1-\gamma}
$$
$$
+ \frac{2\gamma E_{approx,k+1}}{(1-\gamma)^2} + \frac{2\gamma^2 E_{bias,k+1}}{(1-\gamma)^4}
$$
$$
+ \frac{6(1-(1-\gamma_c)\lambda_{min}\alpha)^{\frac{1}{2}[J_{k+1}-(t_\alpha+n+1)]}}{(1-\gamma)^3(1-\gamma_c)^{1/2}\lambda_{min}^{1/2}}
$$
$$
+ \frac{70 L_{k+1}[\alpha(t_\alpha+n+1)]^{1/2}}{\lambda_{min}(1-\gamma_c)(1-\gamma)^3} + \frac{2\gamma\beta}{(1-\gamma)^2}
$$

Simplifying:

$$\mathbb{E}[\|Q_{k+1} - Q^{\pi_{k+1,T}}\|_\infty] \le \gamma^{T(k+1)}\|Q_1 - Q_1^{\pi_0}\|_\infty$$
$$+ \sum_{j=1}^{k+1}\left(\frac{2\gamma E_{approx,j}}{(1-\gamma)^2} + \frac{2\gamma^2 E_{bias,j}}{(1-\gamma)^4}\right.$$
$$+ \frac{6(1-(1-\gamma_c)\lambda_{min}\alpha)^{\frac{1}{2}[J_j-(t_\alpha+n+1)]}}{(1-\gamma)^3(1-\gamma_c)^{1/2}\lambda_{min}^{1/2}}$$
$$\left.+ \frac{70L_j[\alpha(t_\alpha+n+1)]^{1/2}}{\lambda_{min}(1-\gamma_c)(1-\gamma)^3} + \frac{2\gamma\beta}{(1-\gamma)^2}\right)$$
$$+ \sum_{j=1}^{k}\|Q_{k+1} - Q_j\|_{d_{k+1}}$$

where we used the fact that

$$\|Q_{k+1}^* - Q_k^*\|_{d_{k+1}} + \gamma^T \sum_{j=1}^{k-1}\|Q_k^* - Q_j^*\|_{d_k} \le \sum_{j=1}^{k}\|Q_{k+1}^* - Q_j^*\|_{d_{k+1}} \tag{7}$$

due to the triangle inequality and the curriculum design. By induction, for the final curriculum step $K$, we have:

$$\mathbb{E}[\|Q_K - Q^{\pi_K}\|_\infty] \le \gamma^{TK}\|Q_1 - Q_1^{\pi_0}\|_\infty$$
$$+ \sum_{k=1}^{K}\left(\frac{2\gamma E_{approx,k}}{(1-\gamma)^2} + \frac{2\gamma^2 E_{bias,k}}{(1-\gamma)^4}\right.$$
$$+ \frac{6(1-(1-\gamma_c)\lambda_{min}\alpha)^{\frac{1}{2}[J_k-(t_\alpha+n+1)]}}{(1-\gamma)^3(1-\gamma_c)^{1/2}\lambda_{min}^{1/2}}$$
$$\left.+ \frac{70L_k[\alpha(t_\alpha+n+1)]^{1/2}}{\lambda_{min}(1-\gamma_c)(1-\gamma)^3} + \frac{2\gamma\beta}{(1-\gamma)^2}\right)$$
$$+ \sum_{k=1}^{K-1}\|Q_K - Q_k\|_{d_K}$$

This completes the proof.

$\square$

### A.1.2 Proof of Theorem 3.2

*Proof.* From the Finite-Sample Theorem for CRL, we aim to control all error terms to achieve the desired accuracy. We allocate the total error budget $\epsilon$ across the $K$ curriculum steps, with $\epsilon_k$ being the error allocation for step $k$, such that $\sum_{k=1}^{K}\epsilon_k \le \epsilon$. For each curriculum $k$, we need to control the following terms:

$$\gamma^{TK}\|Q_1^* - Q_1^{\pi_0}\|_\infty \le \frac{\epsilon}{4} \tag{8}$$

$$\frac{6(1-(1-\gamma_c)\lambda_{min}\alpha)^{\frac{1}{2}[J_k-(t_\alpha+n+1)]}}{(1-\gamma)^3(1-\gamma_c)^{1/2}\lambda_{min}^{1/2}} \le \frac{\epsilon_k}{4} \tag{9}$$

$$\frac{70L_k[\alpha(t_\alpha+n+1)]^{1/2}}{\lambda_{min}(1-\gamma_c)(1-\gamma)^3} \le \frac{\epsilon_k}{4} \tag{10}$$

$$\frac{2\gamma\beta}{(1-\gamma)^2} \le \frac{\epsilon_k}{4} \tag{11}$$

We can solve each of these constraints. For the first constraint, we need:

$$T \geq \frac{1}{K} \log_\gamma \left( \frac{\epsilon}{4\|Q_1^* - Q_1^{\pi_0}\|_\infty} \right)$$

Since $\|Q_1^* - Q_1^{\pi_0}\|_\infty \leq \frac{1}{1-\gamma}$, we have:

$$T \geq \frac{1}{K} \log \gamma \left( \frac{\epsilon(1-\gamma)}{4} \right) = O\left( \frac{\log(1/\epsilon)}{K} \right)$$

For the second constraint, we need:

$$(1 - (1-\gamma_c)\lambda_{min}\alpha)^{\frac{1}{2}[J_k - (t_\alpha + n + 1)]} \leq \frac{\epsilon_k(1-\gamma)^3(1-\gamma_c)^{1/2}\lambda_{min}^{1/2}}{24},$$

thus by taking log,

$$\frac{1}{2}[J_k - (t_\alpha + n + 1)] \log(1 - (1-\gamma_c)\lambda_{min}\alpha) \leq \log\left( \frac{\epsilon_k(1-\gamma)^3(1-\gamma_c)^{1/2}\lambda_{min}^{1/2}}{24} \right),$$

thus

$$J_k \geq (t_\alpha + n + 1) + \frac{2\log\left( \frac{\epsilon_k(1-\gamma)^3(1-\gamma_c)^{1/2}\lambda_{min}^{1/2}}{24} \right)}{\log(1 - (1-\gamma_c)\lambda_{min}\alpha)}.$$

Using $\log(1-x) \approx -x$ for small $x$, and $\alpha(t_\alpha + n + 1) \leq \frac{(1-\gamma_c)\lambda_{min}}{130 L_k^2}$, we have:

$$J_k \geq (t_\alpha + n + 1) - \frac{2\log\left( \frac{24}{\epsilon_k(1-\gamma)^3(1-\gamma_c)^{1/2}\lambda_{min}^{1/2}} \right)}{(1-\gamma_c)\lambda_{min}\alpha}$$

$$= (t_\alpha + n + 1) + \frac{2\log\left( \frac{\epsilon_k(1-\gamma)^3(1-\gamma_c)^{1/2}\lambda_{min}^{1/2}}{24} \right)}{(1-\gamma_c)\lambda_{min}\alpha}$$

For the third constraint, we need:

$$\alpha(t_\alpha + n + 1) \leq \frac{\epsilon_k^2(1-\gamma)^6(1-\gamma_c)^2\lambda_{min}^2}{4900 L_k^2}$$

$$= O\left( \frac{\epsilon_k^2(1-\gamma)^6(1-\gamma_c)^2\lambda_{min}^2}{L_k^2} \right)$$

For the fourth constraint, we need:

$$\beta \leq \frac{\epsilon_k(1-\gamma)^2}{8\gamma}$$

Combining these constraints, the dominant factor in the sample complexity comes from the third constraint, which gives us:

$$\alpha \leq O\left( \frac{\epsilon_k^2(1-\gamma)^6(1-\gamma_c)^2\lambda_{min}^2}{L_k^2(t_\alpha + n + 1)} \right)$$

The mixing time $t_\alpha$ is $O(\log(1/\alpha))$, which gives us:

$$\alpha \leq O\left( \frac{\epsilon_k^2(1-\gamma)^6(1-\gamma_c)^2\lambda_{min}^2}{L_k^2(\log(1/\alpha) + n + 1)} \right)$$

This implies:

$$\alpha = O\left( \frac{\epsilon_k^2(1-\gamma)^6(1-\gamma_c)^2\lambda_{min}^2}{L_k^2 \log(1/\alpha)} \right)$$

$$\alpha \log(1/\alpha) = O\left( \frac{\epsilon_k^2(1-\gamma)^6(1-\gamma_c)^2\lambda_{min}^2}{L_k^2} \right)$$

Using the Lambert W function, we can solve for $\alpha$:

$$\alpha = \Theta\left(\frac{\epsilon_k^2(1-\gamma)^6(1-\gamma_c)^2\lambda_{min}^2}{L_k^2\log\left(\frac{L_k^2}{\epsilon_k^2(1-\gamma)^6(1-\gamma_c)^2\lambda_{min}^2}\right)}\right)$$

$$= \tilde{\Theta}\left(\frac{\epsilon_k^2(1-\gamma)^6(1-\gamma_c)^2\lambda_{min}^2}{L_k^2}\right)$$

where $\tilde{\Theta}$ hides logarithmic factors. The number of samples required for curriculum step $k$ is:

$$M_k = J_k \cdot T$$

$$= O\left(\left((t_\alpha + n + 1) + \frac{\log(1/\epsilon_k)}{(1-\gamma_c)\lambda_{min}\alpha}\right) \cdot O\left(\frac{\log(1/\epsilon)}{K}\right)\right)$$

$$= O\left(\frac{\log(1/\alpha) + n + 1}{K}\log(1/\epsilon) + \frac{\log(1/\epsilon_k)\log(1/\epsilon)}{K(1-\gamma_c)\lambda_{min}\alpha}\right)$$

$$= O\left(\frac{\log(1/\epsilon)\log\left(\frac{L_k^2}{\epsilon_k^2(1-\gamma)^6(1-\gamma_c)^2\lambda_{min}^2}\right) + n\log(1/\epsilon)}{K}\right.$$

$$\left. + \frac{\log(1/\epsilon_k)\log(1/\epsilon)L_k^2\log\left(\frac{L_k^2}{\epsilon_k^2(1-\gamma)^6(1-\gamma_c)^2\lambda_{min}^2}\right)}{K\epsilon_k^2(1-\gamma)^6(1-\gamma_c)^3\lambda_{min}^3}\right)$$

$$= \tilde{O}\left(\frac{n\log(1/\epsilon)}{K} + \frac{\log(1/\epsilon_k)\log(1/\epsilon)L_k^2}{K\epsilon_k^2(1-\gamma)^6(1-\gamma_c)^3\lambda_{min}^3}\right)$$

The second term dominates, giving us:

$$M_k = \tilde{O}\left(\frac{\log^2(1/\epsilon_k)\log(1/\epsilon)L_k^2}{K\epsilon_k^2(1-\gamma)^6(1-\gamma_c)^3\lambda_{min}^3}\right) \tag{12}$$

The total number of samples across all curriculum steps is:

$$M_{CRL} = \sum_{k=1}^{K} M_k$$

$$= \sum_{k=1}^{K} \tilde{O}\left(\frac{\log^2(1/\epsilon_k)\log(1/\epsilon)L_k^2}{K\epsilon_k^2(1-\gamma)^6(1-\gamma_c)^3\lambda_{min}^3}\right)$$

$$= \tilde{O}\left(\frac{\log(1/\epsilon)}{K}\sum_{k=1}^{K}\frac{\log^2(1/\epsilon_k)L_k^2}{\epsilon_k^2(1-\gamma)^6(1-\gamma_c)^3\lambda_{min}^3}\right)$$

Assuming $\epsilon_k = \frac{\epsilon}{K}$ for all $k$ (uniform error allocation), we get:

$$M_{CRL} = \tilde{O}\left(\frac{\log(1/\epsilon)}{K}\sum_{k=1}^{K}\frac{\log^2(K/\epsilon)L_k^2}{(\epsilon/K)^2(1-\gamma)^6(1-\gamma_c)^3\lambda_{min}^3}\right)$$

$$= \tilde{O}\left(\frac{\log(1/\epsilon)\log^2(K/\epsilon)K}{\epsilon^2(1-\gamma)^6(1-\gamma_c)^3\lambda_{min}^3}\sum_{k=1}^{K}L_k^2\right)$$

$$= \tilde{O}\left(\frac{\log^3(1/\epsilon)K}{\epsilon^2(1-\gamma)^6(1-\gamma_c)^3\lambda_{min}^3}\sum_{k=1}^{K}L_k^2\right)$$

Adding the dependence on the bootstrapping parameter $n$, we have:

$$M_{CRL} = O\left(\sum_{k=1}^{K}\frac{\log^3(1/\epsilon_k)}{\epsilon_k^2} \cdot \tilde{O}\left(\frac{L_k^2 n}{(1-\gamma)^7(1-\gamma_c)^3\lambda_{min}^3}\right)\right) \tag{13}$$

This completes the proof for the first half of the Theorem. Now we move on to prove the second half of the Theorem.

The sample complexity for the final task $K$ is:

$$M_K = O\left(\frac{\log^3(1/\epsilon_K)}{\epsilon_K^2} \cdot L_K^2 \cdot \frac{C}{(1-\gamma)^7(1-\gamma_c)^3 \lambda_{min}^3}\right).$$

We can factor this term out of the total sum:

$$M_{CRL} = M_K \cdot \left(1 + \sum_{k=1}^{K-1} \frac{L_k^2 \cdot \log^3(1/\epsilon_k) \cdot \epsilon_K^2}{L_K^2 \cdot \log^3(1/\epsilon_K) \cdot \epsilon_k^2}\right).$$

Define the curriculum efficiency factor:

$$CEF = 1 + \sum_{k=1}^{K-1} \frac{L_k^2 \cdot \log^3(1/\epsilon_k) \cdot \epsilon_K^2}{L_K^2 \cdot \log^3(1/\epsilon_K) \cdot \epsilon_k^2}.$$

This factor represents the ratio of total curriculum sample complexity to the sample complexity of just the final task. We make two structure assumptions about our curriculum design, geometric error allocation and $L_k$ progression, $\epsilon_k = \epsilon_K \cdot e^{K-k}$ and $L_k = L_K/l^{K-k}$, where $e > 1$. This reflect a curriculum that gradually increases in difficulty while allowing larger errors in earlier stages. For a well-designed curriculum where early tasks are simpler than later ones, $l > 1$.

For the error ratio term:

$$\frac{\epsilon_K^2}{\epsilon_k^2} = \frac{\epsilon_K^2}{(\epsilon_K \cdot e^{K-k})^2} = e^{-2(K-k)},$$

and for the $L_k$ ratio term:

$$\frac{L_k^2}{L_K^2} = \frac{L_k^2}{(L_k \cdot l^{K-k})^2} = l^{-2(K-k)}.$$

For the logarithmic term:

$$\log(1/\epsilon_k) = \log(1/(\epsilon_K \cdot e^{K-k})) = \log(1/\epsilon_K) - (K-k)\log(e).$$

Substituting these expressions into $CEF$:

$$CEF = 1 + \sum_{k=1}^{K-1} \frac{l^{-2(K-k)} \cdot [\log(1/\epsilon_K) - (K-k)\log(e)]^3 \cdot e^{-2(K-k)}}{\log^3(1/\epsilon_K)}.$$

Since $e > 1$ and $K \geq k$, $(K-k)\log(e) \geq 0$. Thus we have

$$CEF = 1 + \sum_{k=1}^{K-1} \frac{[\log(1/\epsilon_K) - (K-k)\log(e)]^3}{\log^3(1/\epsilon_K)} \cdot (el)^{-2(K-k)}$$

$$\leq 1 + \sum_{k=1}^{K-1} (el)^{-2(K-k)}.$$

The sum is a geometric series:

$$\sum_{k=1}^{K-1} (el)^{-2(K-k)} = \sum_{k=1}^{K-1} (el)^{2(k-K)} = (el)^{2(1-K)} \frac{1-(el)^{2(K-1)}}{1-(el)^2} = \frac{(el)^{2(1-K)}-1}{1-(el)^2}$$

CRL is more sample-efficient than direct learning when:

$$M_{CRL} < M_{Direct}$$

Since $M_{CRL} = M_K \cdot CEF$ and for a comparable direct learning approach, $M_{Direct} = M_K \cdot m$ where $m$ represents the relative difficulty factor of direct learning, our condition becomes:

$$CEF < m.$$

Thus for $M_{CRL} < M_{Direct}$, substituting expressions it becomes:

$$M_{CRL} = M_K \cdot CEF \le M_K \cdot (1 + \frac{(el)^{2(1-K)} - 1}{1 - (el)^2}) < M_K \cdot m = M_{Direct}.$$

Thus

$$M_{\text{CRL}} < M_{\text{Direct}} \iff \frac{(el)^{2(1-K)} - 1}{1 - (el)^2} < m - 1.$$

This completes the proof.

$\square$

## B  SAMPLE EFFICIENCY GAINS WITH CRL

Our theoretical analysis (see Sec.3.3) establishes that curriculum reinforcement learning (CRL) attains target performance while requiring fewer hard samples than non-curriculum RL methods. To show this empirically, we count how many training samples from each difficulty level are seen during post-training. All methods are trained for 1600 iterations with an effective batch size of 8, totaling 12,800 samples, which allows a direct comparison of their sample efficiency under the same budget. For simplicity, we perform this analysis on Blocksworld, shown Table 8. Our empirical results show that CRL methods are between $2.5 - 3\times$ more sample efficient than non-CRL methods that train exclusively on hard samples, while also achieving better performance, underscoring the importance of curriculum design for post-training.

Table 8: Consistent with our theoretical guarantees, CRL methods (E2H-C, E2H-G) attain strong performance while requiring substantially fewer hard samples than non-curriculum baselines. For example, E2H-G uses 3580 hard samples versus 12800 for GRPO (HARD). Training exclusively on OOD (GRPO-OOD) trained on 12800 OOD samples performs poorly (see Table 3).

| Method | Number of Training Samples | | | | | |
| --- | --- | --- | --- | --- | --- | --- |
| | Trivial | Easy | Medium | Hard | OOD | Total |
| GRPO (All) | 3200 | 3200 | 3200 | 3200 | 0 | 12800 |
| GRPO (HARD) | 0 | 0 | 0 | 12800 | 0 | 12800 |
| GRPO (OOD) | 0 | 0 | 0 | 0 | 12800 | 12800 |
| E2H-C | 3200 | 3200 | 3200 | 3200 | 0 | 12800 |
| E2H-G (0.5, 0.5) | 1628 | 2997 | 4595 | 3580 | 0 | 12800 |

## C  DATASET DETAILS

In this section we provide details of the datasets used for evaluation. We categorize the datasets into two categories, datasets that contain human annotated difficulties and others that do not.

### C.1  DATASETS WITH HUMAN ANNOTATED DIFFICULTIES

**Blocksworld** (Valmeekam et al., 2023) is a dataset used to evaluate the planning capabilities of LLMs. Each task involves transitioning from an initial block configuration to a target configuration, which requires LLMs to generate a sequence of actions, or plan to achieve the goal. Tasks become more difficult as the required number of steps increases, since the model must reason over longer sequences and maintain correct intermediate states. To study this, we group tasks into four in-distribution difficulty levels: Trivial with 1 step, Easy with 2 steps, Medium with 4 steps, and Hard with 6 steps. Additionally, we include an out-of-distribution (OOD) split with 8-step plans to test generalization

Table 9: Difficulty splits for datasets with human-annotated difficulty levels. Each dataset is categorized based on task-specific properties, specifically, plan length for Blocksworld, number of operands for Countdown, and problem level for MATH.

| Difficulty | Blocksworld (Plan Length) | Countdown (Num. Operands) | MATH (Problem Level) |
|---|---|---|---|
| Trivial (T) | 1 | 2 | 1 |
| Easy (E) | 2 | 3 | 2 |
| Medium (M) | 4 | 4 | 3 |
| Hard (H) | 6 | 5 | 4 |
| OOD | 8 | 6 | 5 |

beyond the training distribution. Trivial tasks are especially simple because the model only needs to predict one correct action out of four possible choices to complete the plan. This setup allows the LLM to first grasp fundamental planning mechanics, which can then be leveraged to learn more complex multi-step tasks.

**Countdown** (Gandhi et al., 2024) is a task where the model must reach a target value by combining given numbers using basic arithmetic operations. While the original dataset uses four numbers per instance, we extend it to create a range of difficulty levels based on the number of input numbers, mainly, Trivial (2), Easy (3), Medium (4), Hard (5), and OOD (6). As the number of inputs increases, the space of possible operation sequences grows rapidly, making it harder for the model to identify the correct combination and order of operations. In contrast, the trivial setting is extremely simple, requiring just one operation between two numbers to reach the target, allowing the model to first learn basic arithmetic before scaling to more complex multi-step problems.

**MATH** (Hendrycks et al., 2021) is a benchmark of 7,500 training and 5,000 test problems covering high-school level mathematics, with each problem labeled from Level 1 (easiest) to Level 5 (hardest). The dataset covers topics such as algebra, geometry, number theory, and probability, including step-by-step solutions. We create a difficulty-based setup using the existing labels, specifically, Trivial (Level 1), Easy (Level 2), Medium (Level 3), Hard (Level 4), and OOD (Level 5). As difficulty increases, problems require more complex reasoning, multi-step solutions, and deeper mathematical understanding. Trivial problems are typically short and rely on basic techniques, making them ideal for teaching foundational reasoning. We use Levels 1 through 4 for training and treat Level 5 as out-of-distribution to assess generalization to the most difficult problems.

We provide the summery of the difficulty splits for each dataset in Table 9.

## C.2 Datasets without Human Annotated Difficulties

**GSM8K** (Cobbe et al., 2021) is a dataset of high-quality, linguistically diverse grade school math word problems designed to evaluate multi-step arithmetic reasoning. Each problem typically requires between two and eight steps involving basic arithmetic operations such as addition, subtraction, multiplication, and division. To assess performance across varying difficulty levels, we create a 4-way split into Trivial, Easy, Medium, and Hard, based on model error rates, as described in Section. 4 and illustrated in Figure. 5

**AQuA** (Ling et al., 2017b) is a dataset of algebraic word problems with multiple-choice answers and detailed rationales, designed to test arithmetic reasoning and symbolic manipulation. For our experiments, we randomly sample 5,000 problems for training. Similar to GSM8K, we use model error rates to define four difficulty levels, Trivial, Easy, Medium, and Hard (see Figure. 6).

Since both GSM8K and AQuA lack explicit difficulty annotations, we construct difficulty splits using model error rates and do not include an out-of-distribution (OOD) category for these datasets.

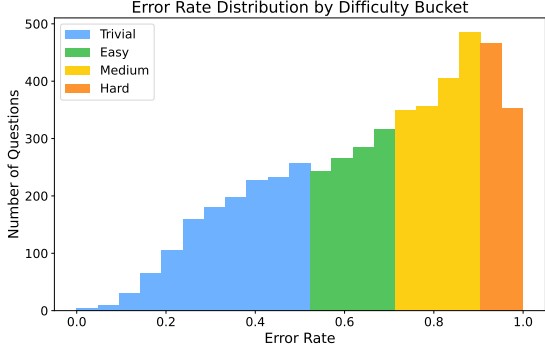

Figure 6: **AQuA** Difficulty distribution based on error rates, grouped into quartiles. Difficulty groups used for training are derived from quartiles of this distribution.

## D    MODEL LICENSE INFORMATION

For our experiments, we use LLaMA 3.2 3B Instruct, Qwen 2.5 1.5B Instruct, and Qwen 2.5 3B Instruct. Qwen 2.5 1.5B Instruct is released under the Apache 2.0 License, Qwen 2.5 3B Instruct under the Qwen Research License, and LLaMA 3.2 3B Instruct under the Meta Community License.

## E    IMPLEMENTATION DETAILS

In this section, we provide implementation details for **E2H Reasoner**. We begin by describing the hardware setup, followed by training parameters, and finally the inference settings used to report the final results.

### E.1    HARDWARE DETAILS

For our experiments, we use up to three 80GB NVIDIA A100 GPUs, particularly for the 3B parameter models. One GPU is dedicated to vLLM for fast inference, while the remaining two are used for model training. On average our experiments require anywhere between 16-18 hours for training.

### E.2    TRAINING DETAILS

We use GRPO as the policy optimization algorithm, combined with parameter-efficient fine-tuning using LoRA. All models are trained for up to 1600 GRPO steps, with hyperparameters listed in Table 10. We adopt the reward design from Guo et al. (2025), where outputs with the correct format receive partial rewards, and full rewards are given only when both format and answer are correct.

For fair comparison, the same configuration is used in our SFT experiments. All training is conducted using the TRL library from Hugging Face, averaged across 3 seeds. For the Self-Evolve baseline (Chen et al., 2025), we use the same hyper-parameters as reported by authors. For DAPO (Yu et al., 2025), we set $n = 4$ on default

### E.3    DIFFICULTY-BASED TRAINING SPLIT CREATION FOR GSM8K AND AQUA

Since both GSM8K and AQuA lack explicit difficulty annotations, we construct difficulty splits based on model error rates. We use CoT prompting with 1-shot in-context learning (ICL) to generate answers for each question. The prompts and ICL examples have been directly adopted from prior works (Hao et al., 2024; Parashar et al., 2025). The model is queried 20 times per question, and the error is computed as the fraction of incorrect responses relative to the ground truth. Based on this, samples with 0–25% error rate are labeled as *trivial*, 25–50% as *easy*, 50–75% as *medium*, and 75–100% as *hard*.

Table 10: Training hyperparameters used for GRPO post-training and LoRA adaptation.

| Component | Parameter | Value |
|---|---|---|
| | learning_rate | 1e-6 |
| | lr_scheduler_type | cosine |
| | per_device_train_batch_size | 2 |
| | gradient_accumulation_steps | 4 |
| | gradient_checkpointing | true |
| GRPO | max_steps | 1600 |
| | bf16 | true |
| | tf32 | true |
| | num_generations | 8 |
| | beta | 0.001 |
| | use_vllm | true |
| | vllm_gpu_memory_utilization | 0.2 |
| | r | 32 |
| | alpha | 64 |
| LoRA | dropout | 0.1 |
| | target_modules | q_proj, v_proj |
| | task_type | CAUSAL_LM |

### E.4 INFERENCE DETAILS

For inference, we use a temperature of 0.0 in all experiments to ensure deterministic outputs and reproducibility. For pass@$k$ evaluations, we enable sampling with a temperature of 0.7, top_p of 0.9, and top_k of 50, where top_k controls the number of candidate tokens considered at each decoding step.

## F COMPARISONS WITH SFT

We compare vanilla supervised fine-tuning (SFT) with RL-based post-training methods (see Table 11). The results show that SFT performance varies significantly across tasks. For instance, in Blocksworld (Valmeekam et al., 2023), where most problems involve fewer than four blocks, SFT performs well—the LLM learns effectively due to the small gap between the training and test distributions. This aligns with prior work suggesting that SFT is most effective when the training data is high quality and closely aligned with the downstream task (Muennighoff et al., 2025; Chu et al., 2025). In contrast, on tasks like Countdown, where the problem distribution is much broader, SFT performs notably worse than RL-based post-training. This gap is especially clear in the out-of-distribution (OOD) setting, where SFT fails to solve any Countdown OOD examples, unlike in Blocksworld, where its performance generalizes more successfully. We highlight these two tasks to illustrate how SFT performance can vary across distributions, in contrast to the more consistent behavior of RL-based post-training methods.

## G ADDITIONAL ANALYSIS AND ABLATIONS

### G.1 RESULTS ON MORE LLMS

Additional results on more LLMs are reported in this subsection.

### G.2 RESULTS ACROSS GAUSSIAN PARAMETERS

In this section, we expand on the results presented in the main paper by reporting all three parameter settings of Gaussian scheduling in Table 13. Similarly, for Qwen 2.5 1.5B Instruct we include the results for all Gaussian scheduling variants on GSM8K and AQuA in Table 14.

Table 11: Comparison of RL-based post-training and supervised fine-tuning (SFT). SFT performance varies noticeably across tasks, highlighting its inconsistency in generalizing across domains.

| Models | Methods | Blocksworld | | | | | Countdown | | | | |
|---|---|---|---|---|---|---|---|---|---|---|---|
| | | Trivial | Easy | Med | Hard | OOD | Trivial | Easy | Med | Hard | OOD |
| Qwen 2.5 / 1.5B Instruct | SFT | 100.0 | 97.8 | 88.1 | 55.3 | 16.5 | 97.4 | 41.8 | 14.2 | 4.6 | 0.0 |
| | GRPO | 98.0 | 100.0 | 84.5 | 26.3 | 5.3 | 96.1 | 64.9 | 28.8 | 18.1 | 9.2 |
| | E2H-G | 98.0 | 100.0 | 95.3 | 32.9 | 7.3 | 95.2 | 84.1 | 48.1 | 28.1 | 14.2 |
| | E2H-C | 100.0 | 100.0 | 15.5 | 0.0 | 0.0 | 96.7 | 64.0 | 25.9 | 15.8 | 6.4 |
| Qwen 2.5 / 3B Instruct | SFT | 100.0 | 100.0 | 94.0 | 62.5 | 27.8 | 99.9 | 63.9 | 30.8 | 13.3 | 0.0 |
| | GRPO | 100.0 | 100.0 | 88.1 | 38.8 | 14.6 | 99.6 | 89.3 | 73.4 | 41.7 | 9.4 |
| | E2H-G | 100.0 | 100.0 | 96.4 | 53.3 | 23.2 | 98.5 | 90.8 | 71.0 | 43.5 | 19.4 |
| | E2H-C | 100.0 | 100.0 | 94.1 | 52.0 | 22.5 | 100.0 | 90.4 | 69.7 | 35.7 | 12.6 |
| LLaMA 3.2 / 3B Instruct | SFT | 100.0 | 97.8 | 96.4 | 72.3 | 37.8 | 100.0 | 66.6 | 27.2 | 12.1 | 0.0 |
| | GRPO | 100.0 | 100.0 | 94.1 | 38.9 | 13.3 | 99.9 | 89.5 | 71.6 | 47.9 | 2.7 |
| | E2H-G | 100.0 | 100.0 | 98.8 | 44.1 | 17.6 | 95.0 | 89.9 | 73.3 | 46.5 | 24.3 |
| | E2H-C | 100.0 | 0.0 | 0.0 | 0.0 | 0.0 | 100.0 | 55.3 | 0.0 | 0.0 | 0.0 |

Table 12: Results on Qwen-2.5 3B Instruct.

| | Blocksworld | | | | | Countdown | | | | | MATH | | | | |
|---|---|---|---|---|---|---|---|---|---|---|---|---|---|---|---|
| | Trivial | Easy | Med | Hard | OOD | Trivial | Easy | Med | Hard | OOD | Trivial | Easy | Med | Hard | OOD |
| | | | | | | Qwen 2.5 3B Instruct | | | | | | | | | |
| CoT | 1.0 | 6.7 | 7.1 | 1.0 | 0.0 | 24.0 | 15.9 | 4.4 | 0.5 | 0.0 | 68.0 | 55.8 | 48.8 | 33.4 | 20.0 |
| GRPO (All) | **100** | **100** | 88.1 | 38.3 | 14.6 | 99.6 | 89.3 | **73.4** | 41.7 | 9.4 | 90.1 | 78.5 | 71.9 | 57.4 | 36.8 |
| GRPO (Hard) | 50.0 | 42.2 | 86.9 | **72.4** | **48.3** | 1.6 | 49.5 | 38.7 | 27.7 | 12.6 | 90.6 | 78.4 | 70.5 | 58.0 | 36.0 |
| GRPO (OOD) | 0.0 | 13.3 | 9.5 | 0.0 | 0.0 | 0.4 | 25.1 | 21.2 | 13.8 | 8.5 | 91.3 | 78.1 | 70.0 | 58.7 | 36.8 |
| CL | **100** | **100** | 77.3 | 42.1 | 20.5 | 39.0 | 91.0 | 72.9 | 38.7 | 12.5 | 90.6 | 78.9 | 72.5 | 57.2 | 36.6 |
| Self-Evolve | **100** | **100** | 88.1 | 42.5 | 18.5 | 99.6 | **91.4** | 79.8 | 56.8 | 29.3 | **92.2** | 80.8 | 70.6 | 58.6 | 37.1 |
| E2H-G | **100** | **100** | **96.4** | 53.3 | 23.2 | 98.5 | 90.8 | **81.2** | **60.7** | **32.3** | 90.4 | **81.0** | 71.5 | 58.2 | 37.2 |
| E2H-C | **100** | **100** | 94.1 | 52.0 | 22.5 | **100** | 90.4 | 69.7 | 35.7 | 12.6 | 90.4 | 80.0 | **73.4** | **58.9** | **38.1** |

## G.3 CHOICE OF NUMBER OF DIFFICULTY SPLITS

The number of difficulty splits can be treated as a hyperparameter in **E2H Reasoner**, particularly for datasets without human-annotated difficulty labels such as GSM8K (Cobbe et al., 2021) and AqUA (Ling et al., 2017a). We set this value to 4, consistent with the range of 3 to 5 used in the curriculum learning literature (Bengio et al., 2009). As shown in Table 15, our method remains robust across different choices of this parameter.

## G.4 DAPO + E2H

Combining E2H with DAPO consistently lowers the fraction of advantage-zero batches (see Fig. 7), indicating that the curriculum helps select more informative batches with difficulty better aligned to the model's competence.

## G.5 QUALITATIVE ANALYSIS OF SCHEDULING TECHNIQUES

We provide a qualitative analysis of scheduling techniques in Table 16, highlighting their respective strengths and weaknesses. The results show that different schedulers are suited to different tasks and model scales. Inspired by the effectiveness of **E2H Reasoner**, we hope this motivates the community to explore more sophisticated scheduling strategies for LLM post-training.

Table 13: Effect of scheduling strategies in LLM post-training. We compare balanced scheduling, traditional curriculum learning (CL), and our proposed **E2H Reasoner** variants, namely, E2H-G and E2H-C. CoT is reported as a reference.

| | Blocksworld | | | | | Countdown | | | | | MATH | | | | |
|---|---|---|---|---|---|---|---|---|---|---|---|---|---|---|---|
| | Trivial | Easy | Med | Hard | OOD | Trivial | Easy | Med | Hard | OOD | Trivial | Easy | Med | Hard | OOD |
| Qwen 2.5 3B Instruct | | | | | | | | | | | | | | | |
| CoT | 1.0 | 6.7 | 7.1 | 1.0 | 0.0 | 24.0 | 15.9 | 4.4 | 0.5 | 0.0 | 68.0 | 55.8 | 48.8 | 33.4 | 20.0 |
| Balanced | 100.0 | 100.0 | 88.1 | 38.8 | 14.6 | 99.6 | 86.9 | 57.8 | 23.6 | 7.8 | 90.1 | 78.5 | 71.9 | 57.4 | 36.8 |
| CL | 100.0 | 100.0 | 77.3 | 42.1 | 20.5 | 39.0 | 91.0 | 72.9 | 38.7 | 12.5 | 90.6 | 78.9 | 72.5 | 57.2 | 36.6 |
| E2H-G (0.25, 0.75) | 100.0 | 100.0 | 96.4 | 53.3 | 23.2 | 98.5 | 90.8 | 71.0 | 43.5 | 19.4 | 90.4 | 81.0 | 71.5 | 58.2 | 37.2 |
| E2H-G (0.5, 0.5) | 100.0 | 100.0 | 89.2 | 32.9 | 10.6 | 99.6 | 89.3 | 73.4 | 41.7 | 9.4 | 91.1 | 78.2 | 70.5 | 56.6 | 35.0 |
| E2H-G (0.75, 0.25) | 100.0 | 100.0 | 42.9 | 3.9 | 2.0 | 100.0 | 87.8 | 52.3 | 16.7 | 6.5 | 90.0 | 79.3 | 70.6 | 57.1 | 35.9 |
| E2H-C | 100.0 | 100.0 | 94.1 | 52.0 | 22.5 | 100.0 | 90.4 | 69.7 | 35.7 | 12.6 | 90.4 | 80.0 | 73.4 | 58.9 | 38.1 |
| LLaMA 3.2 3B Instruct | | | | | | | | | | | | | | | |
| CoT | 24.0 | 0.0 | 1.2 | 1.0 | 0.0 | 37.1 | 4.6 | 0.3 | 0.0 | 0.0 | 65.9 | 44.6 | 35.2 | 24.1 | 13.6 |
| Balanced | 100.0 | 100.0 | 94.1 | 38.9 | 13.3 | 99.9 | 89.5 | 71.6 | 47.9 | 2.7 | 65.9 | 47.0 | 36.0 | 22.0 | 10.2 |
| CL | 100.0 | 0.0 | 0.0 | 0.0 | 0.0 | 17.2 | 36.0 | 22.7 | 11.2 | 4.1 | 74.1 | 54.1 | 43.9 | 28.0 | 12.5 |
| E2H-G (0.25, 0.75) | 100.0 | 100.0 | 98.8 | 44.1 | 17.6 | 95.0 | 89.9 | 73.3 | 46.5 | 24.3 | 78.7 | 58.4 | 46.4 | 32.3 | 14.5 |
| E2H-G (0.5, 0.5) | 100.0 | 100.0 | 88.1 | 26.3 | 9.6 | 94.4 | 86.7 | 65.1 | 27.9 | 2.3 | 66.1 | 45.0 | 36.2 | 24.4 | 11.4 |
| E2H-G (0.75, 0.25) | 100.0 | 97.8 | 75.0 | 21.1 | 4.0 | 98.9 | 89.4 | 48.6 | 0.7 | 0.0 | 63.8 | 48.2 | 35.6 | 24.0 | 8.6 |
| E2H-C | 100.0 | 100.0 | 15.5 | 0.0 | 0.0 | 96.7 | 64.0 | 25.9 | 15.8 | 6.4 | 84.6 | 69.6 | 63.0 | 47.6 | 28.6 |

Table 14: Expanded results for Qwen 2.5 1.5B Instruct showing all E2H-G (Gaussian scheduling) variants on GSM8K and AQuA. We provide a comparison against Balanced and E2H-C (Cosine scheduling).

| | AQuA | | | | | GSM8K | | | | |
|---|---|---|---|---|---|---|---|---|---|---|
| | Trivial | Easy | Med | Hard | Avg | Trivial | Easy | Med | Hard | Avg |
| Balanced | 95.8 | 68.0 | 48.6 | 21.0 | 63.3 | 99.0 | 95.3 | 84.1 | 49.9 | 77.1 |
| E2H-C | 86.1 | 72.2 | 48.6 | 26.3 | 62.5 | 98.0 | 95.3 | 83.9 | 46.6 | 75.7 |
| E2H-G (0.25, 0.75) | 93.0 | 70.8 | 40.2 | 23.6 | 61.4 | 98.5 | 96.3 | 83.6 | 51.2 | 77.6 |
| E2H-G (0.5, 0.5) | 90.2 | 81.9 | 43.0 | 34.2 | 66.1 | 97.6 | 94.7 | 89.0 | 51.8 | 78.7 |
| E2H-G (0.75, 0.25) | 83.3 | 79.1 | 54.5 | 26.3 | 64.1 | 98.0 | 97.3 | 85.9 | 47.7 | 77.1 |

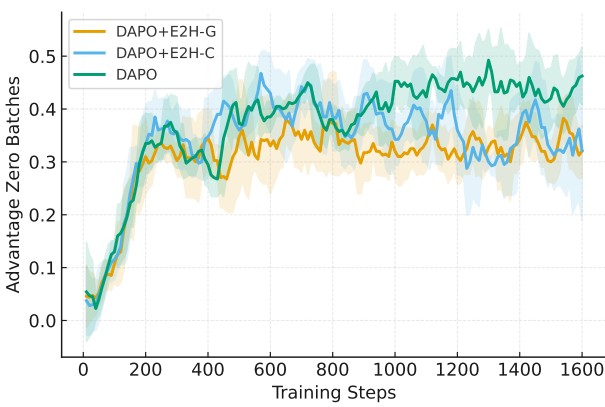

Figure 7: Fraction of advantage-zero batches over training steps for DAPO, DAPO+E2H-G, and DAPO+E2H-C.

## H    LLM USAGE

LLMs were used for text-refining purposes only.

Table 15: Results on GSM8K (Cobbe et al., 2021) across different Gaussian parameter splits. The main experiments use the 4-split setting; below we show that our method is robust to the number of splits.

| Method | 3 Splits | 4 Splits | 5 Splits |
|--------|----------|----------|----------|
| E2H-C | 79.1 | 75.7 | 76.1 |
| E2H-G | 78.6 | 78.7 | 78.5 |

Table 16: Qualitative comparison of scheduling strategies, outlining their strengths and weaknesses in the context of LLM post-training.

| Scheduling | Strengths | Weaknesses |
|------------|-----------|------------|
| Balanced | 1. Default in LLM post-training. 2. Parameter-free and easy to use. | 1. Assumes uniform difficulty across the dataset. 2. Often fails to improve reasoning on hard tasks. |
| Traditional | 1. Simple and parameter-free. 2. Easy to implement. | 1. Can cause overfitting to easy tasks. 2. May lead to forgetting earlier tasks. |
| E2H-C | 1. Parameter-free and simple to apply. 2. Suitable for tasks with similar zero-shot performance across difficulty levels. | 1. May overfit to easy tasks when rewards for hard tasks are sparse. |
| E2H-G | 1. Effective across tasks and models. 2. Enables fine-grained control. | 1. Requires tuning two hyperparameters. |

## I   LIMITATIONS

Despite the strengths of **E2H Reasoner**, it has certain limitations. Our approach uses simple and intuitive probabilistic schedulers, specifically based on Gaussian and cosine functions, which do not adapt during training. While these choices are effective, our results suggest that incorporating adaptive strategies, such as advantage-based scheduling, could offer further improvements. Comparisons with adaptive curriculum methods that aim to maximize learnability reveal an important insight: maximizing learnability does not always lead to stronger reasoning. This outcome depends on the structure and difficulty of the problems within the dataset. Combining our method with adaptive approaches presents a promising direction for future work.

## J   SOCIETAL IMPACTS

Our work introduced **E2H Reasoner**, a curriculum-based reinforcement learning (RL) approach designed to enhance the reasoning capabilities of small-scale large language models (LLMs). This advancement holds significant societal implications. Enhanced reasoning in LLMs can improve decision-making processes in critical domains such as healthcare, education, and legal systems, where nuanced understanding is paramount. Moreover, by empowering smaller models, **E2H Reasoner** promotes broader accessibility to advanced AI capabilities. Although our tasks focus on reasoning, future extensions involving real-world interaction could pose risks, including potential misuse of language models.

