# OpenReview forum: "Curriculum Reinforcement Learning from Easy to Hard Tasks Improves LLM Reasoning"
_ICLR.cc/2026/Conference — ICLR 2026 Poster_

### Official Review · Reviewer_rLJH · 2025-10-26

**Soundness:** 3
**Presentation:** 2
**Contribution:** 2
**Rating:** 4
**Confidence:** 3

**Summary:**

This paper introduces the E2H Reasoner, a reinforcement learning (RL) method for large language models (LLMs) inspired by curriculum learning. The method schedules tasks from easy to hard during training, aiming to accelerate LLM learning. Theoretical analysis shows that curriculum-based reinforcement learning (CRL) requires fewer total samples than directly training on the final task, and experimental results are generally positive.

**Strengths:**

1. The paper provides theoretical justification for why CRL can achieve sample efficiency, requiring fewer total samples than direct learning on the final task.
2. The experimental results are sound and well-presented.

**Weaknesses:**

1. The idea of using curriculum learning to improve RL efficiency is not novel. The paper acknowledged prior work—e.g., Chen et al., Foster et al., Bae et al., Zeng et al. which used curriculum learning ideas. The paper should also cite Yu et al. (DAPO: An Open-Source LLM Reinforcement Learning System at Scale).
2. In the experimental results, E2H does not consistently outperform baselines such as GRPO or Self-Evolve.
3. The paper does not clearly articulate the advantages of E2H over adaptive filtering methods such as DAPO or Self-Evolve. In fact, adaptive filtering—where pass rate determines sample filtering—has several appealing properties:
1) Model-dependent difficulty: “Easy” and “hard” samples are relative to the specific model; what is easy for one model may be hard for another. Thus, classifying samples by difficulty a priori can be problematic.
2) Lack of synchronization: Without adaptive scheduling, the scheduler and the model may become misaligned—for instance, the scheduler may advance to harder tasks before the model is ready.

**Questions:**

See weakness

---

> ### Author Response · Authors · 2025-11-21
> **Response to Reviewer rLJH 1/2**
>
> We thank the reviewer for their positive feedback. They highlight the “**theoretical justification**” provided in our paper for the sample efficiency of Curriculum Reinforcement Learning methods. They also appreciate that the “**experimental results are sound and well-presented**”.
>
> Below, we address all the questions and comments raised by the reviewer. We have also modified our manuscript according to these suggestions. Please note that all changes in the revised manuscript are highlighted in cyan and italicized for clarity.
>
> > Q1. *On the comparison with DAPO*
>
> Thank you for raising this concern. Based on your suggestion, we have added the comparison with DAPO to our revised paper in **Table 5 and 6** for Qwen 1.5B Instruct, and we plan to include results for other models in the final version of the paper. We observe that E2H outperforms DAPO in overall performance.
>
> Additionally, we conduct an ablation where we apply the E2H curriculum on top of DAPO’s adaptive sampling and find consistent gains across all benchmarks. This highlights the complementary nature of the two approaches: DAPO filters out tasks that are too easy or too hard, while the E2H uses probabilistic difficulty-based scheduling to gradually emphasize harder task groups, together leading to improved performance across tasks.
>
> As shown in Tables 5 and 6, DAPO \+ E2H consistently outperforms all baselines on all benchmarks, especially on the harder difficulties. These results highlight that E2H and DAPO are **orthogonal** to each other and can be combined with minimal effort for the best results. We explain this in detail in the revised manuscript (Line 509).
>
> Due to space constraints, we do not reproduce Table 5 here; however, we include Table 6 below for reference.
>
> | Method | Variant | AQUA Triv. | AQUA Easy | AQUA Med. | AQUA Hard | AQUA Avg | GSM8K Triv. | GSM8K Easy | GSM8K Med. | GSM8K Hard | GSM8K Avg |
> |--------|---------|------------|-----------|-----------|-----------|----------|-------------|------------|------------|------------|-----------|
> | GRPO   | Baseline | 95.8 | 68.0 | 48.6 | 21.0 | 63.3 | 99.0 | 95.0 | 84.1 | 49.9 | 77.1 |
> |    | E2H-G    | 90.2 | 81.9 | 43.0 | 34.2 | **66.1** | 97.6 | 94.7 | 89.0 | 51.8 | **78.7** |
> |    | E2H-C    | 86.1 | 72.2 | 48.6 | 26.3 | 62.5 | 98.0 | 95.3 | 83.9 | 46.6 | 75.7 |
> | DAPO   | Baseline | 94.4 | 72.2 | 48.6 | 31.5 | 65.7 | 97.0 | 98.0 | 87.5 | 52.0 | 79.0 |
> |    | E2H-G    | 95.8 | 81.9 | 51.3 | 28.9 |  69.3 | 99.1 | 97.3 | 89.3 | 53.2 | **80.1** |
> |    | E2H-C    | 95.8 | 75.0 | 55.5 | 36.8 | **69.6** | 98.5 | 97.0 | 88.1 | 48.8 | 78.0 |
>
>
> > Q2. *On model-dependent difficulty*
>
> For E2H, we group the datasets (without human-annotated difficulties) into difficulty-based bins separately for each model in a single pre-processing step before training. We detail this procedure in Section A.3 of the Appendix in the revised manuscript. Although models differ in abilities, in practice, reasoning tasks have an inherent notion of difficulty. Our classification of examples by difficulty is useful, as shown by the improved performance of E2H when using this curriculum.
>
> > Q3. *On lack of synchronization*
>
> Computing model-specific difficulties a priori and grouping tasks into bins gives each model a reasonable curriculum. While post-training, if this curriculum were badly unsynchronized with the model, E2H would not yield the consistent gains we observe across tasks and models. Moreover, as shown in Tables 5 and 6 (see our response to Q1), E2H further improves performance when combined with DAPO. Figure 7 in the revised manuscript provides further evidence: combining DAPO with E2H reduces the fraction of training batches with zero advantage. This is because E2H builds the difficulty-aware sampling distribution that DAPO resamples from, leading to the **strongest performance** across benchmarks.
>
> > Q4. *On the comparison with Self-Evolve*
>
> The primary advantage of E2H over Self-Evolve lies in its simplicity of implementation. Self-Evolve involves multiple task and model-specific parameters that require careful tuning. In contrast, E2H-C is completely parameter-free, and E2H-G uses only two parameters, which we ablate across three fixed values shared across all tasks and models. Despite this simplicity, E2H consistently outperforms Self-Evolve, particularly on the hardest difficulty settings, please see our experiment results summarized in Table 4\. Moreover, while we provide a detailed theoretical analysis of our E2H method, the Self-Evolve paper (Chen et al, 2025\) does not provide any such theoretical analysis.

---

> ### Author Response · Authors · 2025-11-21
> **Response to Reviewer rLJH 2/2**
>
> > Q5. *On Curriculum learning for RL and novelty*
>
> While curriculum learning has been addressed in the broader RL literature, its potential for improving the reasoning capabilities of LLM has just started getting attention (mostly starting from the beginning of 2025).  Please note that we have summarized the relevant recent works in our literature review, see Section 2, lines 107-117. Our contributions differ significantly from those of existing works, both in the algorithmic and theoretical contributions.  From an algorithmic perspective, we propose a novel adaptive curriculum scheduling (E2H)  to enable small-scale LLMs to solve challenging reasoning problems that were not possible only via using standard GRPO/DAPO style training. We demonstrate the superior performance of our E2H approach through extensive experiments using multiple LLM models on multiple challenging tasks. From a theoretical perspective, we provide guarantees for the convergence of E2H and improved sample efficiency with respect to the direct RL approach. We sincerely believe that these contributions are original and relevant for the LLM research community.
>
>
> > Q6. *On concerns over E2H performance*
>
> We highlight that E2H consistently outperforms baselines as task difficulty increases. Learning harder tasks in the reinforcement learning with verifiable reward (RLVR) setting is known to be challenging due to the sparse rewards problem [1,2,3,4]. E2H not only improves performance on these harder tasks but also shows stronger generalization, achieving better out-of-distribution performance. This aligns with our definition of reasoning as generalization and reinforces the primary goal of our work, as outlined in Line 31 of the introduction and appreciated by Reviewer  kbdX.
>
> References:
>
> [1] DeepSeekMath: Pushing the Limits of Mathematical Reasoning in Open Language Models, arxiv, 2024.
>
> [2] DeepSeek-R1: Incentivizing Reasoning Capability in LLMs via Reinforcement Learning, Nature, 2025.
>
> [3] SimpleRL-Zoo: Investigating and Taming Zero Reinforcement Learning for Open Base Models in the Wild, COLM, 2025.
>
> [4] DeepSeek-R1: Incentivizing Reasoning Capability in LLMs via Reinforcement Learning, Nature, 2025.

---

> ### Author Response · Authors · 2025-11-27
>
> Dear Reviewer rLJH,
>
> As the discussion phase comes to a close, we sincerely thank you for your time and thoughtful feedback. We have carefully considered your comments and provided detailed responses to each of your concerns in our earlier reply.
>
> We greatly appreciate your assessment of our work, and please don’t hesitate to let us know if there are any remaining questions or points we can clarify.
>
> Warm regards,
>
> Authors

---

### Official Review · Reviewer_mKj1 · 2025-10-31

**Soundness:** 3
**Presentation:** 3
**Contribution:** 3
**Rating:** 6
**Confidence:** 4

**Summary:**

This paper proposes using a curriculum learning approach where they schedule tasks from Easy to Hard during RLVR which shows better performance at the end of training. They provide convergence guarantees for this algorithm in an approximate policy iteration framework and derive finite-sample complexity bounds which show the this is more sample efficient than training without any curriculum.

**Strengths:**

The paper proposes a simple method of using curriculum learning. The curriculum implicitly assumes some grouping of tasks, but they also  show that the grouping is not necessary because tasks can be clustered just using pass rates of the initial model. They also compare with different baselines and the empirical results seem sound.

**Weaknesses:**

The only weakness that comes to mind is not comparing with DAPO [1] which also has an implicit curriculum because the model keeps filtering prompts that are either too easy or too hard. Could the authors compare with DAPO as well and show results on the benchmarks?

Also the paper doesn't cite Paprika [2] which also proposes a curriculum when tasks can be grouped.


[1] DAPO: An Open-Source LLM Reinforcement Learning System at Scale (https://arxiv.org/abs/2503.14476)

[2] Training a Generally Curious Agent (https://arxiv.org/abs/2502.17543)

**Questions:**

Please look at the weaknesses.

---

> ### Author Response · Authors · 2025-11-21
> **Response to Reviewer mKj1**
>
> We thank the reviewer for their positive evaluation of our work. They highlight the “**simple and practical design,**” and appreciate that the paper includes experiments with both predefined groupings and automatically induced curricula based on initial pass rates. The reviewer appreciates the “**improved convergence behavior**” and “**theoretical rigor**” of our work, citing our “**convergence guarantees**” and that it is “**more sample efficient**” than training without any curriculum. Finally, they acknowledge the “**sound**” empirical results in our work.
>
> Below, we address all the questions and comments raised by the reviewer. We have also modified our manuscript according to these suggestions. Please note that all changes in the revised manuscript are highlighted in cyan and italicized for clarity.
>
> > Q1. *Could the authors compare with DAPO?*
>
> Thank you for raising this concern. Based on your suggestion, we have added the comparison with DAPO to our revised paper in **Table 5 and 6** for Qwen 1.5B Instruct, and we plan to include results for other models in the final version of the paper. We observe that E2H outperforms DAPO in overall performance.
>
> Additionally, we conduct an ablation where we apply the E2H curriculum on top of DAPO’s adaptive sampling and find consistent gains across all benchmarks. This highlights the complementary nature of the two approaches: DAPO filters out tasks that are too easy or too hard, while the E2H uses probabilistic difficulty-based scheduling to gradually emphasize harder task groups, together leading to improved performance across tasks.
>
> As shown in Tables 5 and 6, DAPO \+ E2H consistently outperforms all baselines on all benchmarks, especially on the harder difficulties. These results highlight that E2H and DAPO are orthogonal to each other and can be combined with minimal effort for the best results. We explain this in detail in the revised manuscript (Line 509).
>
> Due to space constraints, we do not reproduce Table 5 here; however, we include Table 6 below for reference.
>
> | Method | Variant | AQUA Triv. | AQUA Easy | AQUA Med. | AQUA Hard | AQUA Avg | GSM8K Triv. | GSM8K Easy | GSM8K Med. | GSM8K Hard | GSM8K Avg |
> |--------|---------|------------|-----------|-----------|-----------|----------|-------------|------------|------------|------------|-----------|
> | GRPO   | Baseline | 95.8 | 68.0 | 48.6 | 21.0 | 63.3 | 99.0 | 95.0 | 84.1 | 49.9 | 77.1 |
> |    | E2H-G    | 90.2 | 81.9 | 43.0 | 34.2 | **66.1** | 97.6 | 94.7 | 89.0 | 51.8 | **78.7** |
> |    | E2H-C    | 86.1 | 72.2 | 48.6 | 26.3 | 62.5 | 98.0 | 95.3 | 83.9 | 46.6 | 75.7 |
> | DAPO   | Baseline | 94.4 | 72.2 | 48.6 | 31.5 | 65.7 | 97.0 | 98.0 | 87.5 | 52.0 | 79.0 |
> |    | E2H-G    | 95.8 | 81.9 | 51.3 | 28.9 |  69.3 | 99.1 | 97.3 | 89.3 | 53.2 | **80.1** |
> |    | E2H-C    | 95.8 | 75.0 | 55.5 | 36.8 | **69.6** | 98.5 | 97.0 | 88.1 | 48.8 | 78.0 |
>
>
> > Q2. *On missing references for DAPO and Paprika*
>
> Thanks for the suggestion. We have added the citations to these works in the third paragraph of the related work section in our revised manuscript.

---

> > ### Author Response · Authors · 2025-11-27
> >
> > Dear Reviewer mKj1,
> >
> > As the discussion phase comes to a close, we sincerely thank you for your time and thoughtful feedback. We have carefully considered your comments and provided detailed responses to each of your concerns in our earlier reply.
> >
> > We greatly appreciate your assessment of our work, and please don’t hesitate to let us know if there are any remaining questions or points we can clarify.
> >
> > Warm regards,
> >
> > Authors

---

### Official Review · Reviewer_kbdX · 2025-11-02

**Soundness:** 3
**Presentation:** 3
**Contribution:** 3
**Rating:** 6
**Confidence:** 2

**Summary:**

This paper proposes E2H Reasoner, a curriculum reinforcement learning (CRL) approach for enhancing LLM reasoning capabilities. It decomposes complex tasks into easier subtasks, uses probabilistic schedulers (cosine and Gaussian) to gradually shift from easy to hard tasks during RL post-training, and provides empirical improvements on benchmarks like Blocksworld, Countdown, and arithmetic tasks, achieving SOTA results. Theoretically, it analyzes CRL via approximate policy iteration, proving convergence and reduced sample complexity compared to direct learning.

**Strengths:**

1. The method creatively combines task decomposition with probabilistic scheduling in CRL, addressing rollout inefficiencies in difficult reasoning tasks by building skills incrementally, which makes intuitive sense and extends prior RL post-training like DeepSeek-R1.

2. Theoretical analysis provides finite-sample bounds and convergence guarantees, grounding the approach in approximate policy iteration.

3. Well-structured presentation with illustrative figures (e.g., task decomposition in Fig. 2, schedulers in Figs. 3-4) and precise definitions of reasoning as generalization; methods and experiments are logically sequenced.

**Weaknesses:**

1. Risk of Overfitting in Task Decomposition: Decomposing hard tasks into varying difficulty levels may cause repeated exposure to similar knowledge patterns across subtasks, increasing overfitting risks, especially if subtasks overlap significantly without explicit regularization.


2. Lack of Implementation Details for Reproducibility: Key details are missing, such as prompts used for automatic difficulty estimation (e.g., in AQuA/GSM8K) or exact hyperparameters for task grouping, raising concerns about replicating the reported effects.


3. Limited Scope of Experiments: Evaluations focus on relatively lower-difficulty tasks like Blocksworld and arithmetic benchmarks; lacks experiments on highly challenging ones like AIME, LCB, or agent-based tasks, limiting evidence for broader applicability.

**Questions:**

1. Advantages of Task Decomposition Over Traditional Curriculum Learning: What specific advantages does your task decomposition offer compared to standard curriculum learning (e.g., fixed-stage switching)? Is there a theoretical comparison on generalization, perhaps extending your API framework?


2. Inconsistencies in Model Trends in Figure 1(a): In Figure 1(a) for Countdown, why do Qwen 1.5B and LLaMA 3.2 3B show inconsistent relative performance trends under E2H vs. base models (e.g., one benefits more at low k)? Could this relate to architectural differences or training artifacts?

---

> ### Author Response · Authors · 2025-11-21
> **Response to Reviewer kbdX 1/2**
>
> We thank Reviewer kbdX for appreciating our work as an **“intuitive extension”** to prior RL post-training like DeepSeek-R1. We are glad the reviewer found value in our theoretical analysis and convergence guarantees, **“grounding the work in approximate policy iteration”.** We also appreciate the reviewer’s positive comments on the writing, particularly the **“illustrative figures”,** the **“definitions of reasoning as generalization”,** and that the **“methods and experiments are logically sequenced.”**
>
> Below, we address all the questions and comments raised by the reviewer. We have also modified our manuscript according to these suggestions. Please note that all changes in the revised manuscript are highlighted in cyan and italicized for clarity.
>
> > Q1. *On the Confusion between dataset-level task decomposition and subtask creation*
>
> We thank the reviewer for raising this concern. However, as clarified in Line 144 of Section 3.1, our decomposition is applied at the dataset level, where we group existing training examples based on **difficulty**, not by breaking a single hard problem into subtasks. This means the issue of overfitting due to overlapping sub-problems **does not** apply. Instead, our method gradually progresses through tasks of increasing difficulty using the E2H curriculum To further clarify this point, we have revised the caption of Figure 2 and reiterated the goal of our approach in Line 169.
>
> > Q2. *On the request for details on prompts used for difficulty estimation and hyperparameters.*
>
> We thank the reviewer for pointing out the need for more detail. Since GSM8K and AQuA lack explicit difficulty annotations, we estimate per-question error rates by querying each model 20 times using CoT prompting with 1-shot ICL examples. These prompts are adapted from prior work \[1,2\]. Based on the fraction of incorrect responses, we assign difficulty levels: 0–25% error as *trivial*, 25–50% as *easy*, 50–75% as *medium*, and 75–100% as *hard*. This process is used to construct our curriculum-based training splits.
>
> To clarify this in the paper:
>
> * We revised the captions of Figures 5 and 6 to mention the quartile-based difficulty grouping explicitly. We also add an explanation about this in Section 3.1 in Line 167.
>
> * We added a new subsection in the Appendix (Section A.3) titled “*Difficulty-Based Training Split Creation for GSM8K and AQuA”* with a detailed explanation.
>
> * We add details about the reward formulation in the training details outlined in the Appendix Section E.2.
>
> For further inspection, all implementation details are included in our anonymized code repository https://anonymous.4open.science/r/E2H-Reasoning-1F19
>
> > Q3. *Request for evaluation on more challenging tasks, such as AIME*
>
> Thank you for this suggestion. We extend the evaluation of our models trained on MATH to two challenging datasets, AIME and OlympiadBench \[3\]. E2H achieves the strongest performance across both, with results reported in terms of accuracy and pass@32 below.. We include these results as part of the main paper in Table 5, and add an explanation in Line 485.
>
> | Model       | Method        | AIME24 |   AIME24@32 |  OlympiadBench | OlympiadBench @32|
> |-------------|---------------|--------|----------------|-------|-------|
> | Qwen 1.5B   | GRPO          | 3.3  | 10.0   | 7.3   | 33.3 |
> | Qwen 1.5B   | Self Evolve   | 3.3 | 10.0    | 8.0     | 37.0 |
> | Qwen 1.5B   | E2H-G         | 6.7 | 16.67   | 8.7  | 39.3 |
> | Qwen 1.5B   | E2H-C         | 6.7 | 16.67   | 10.0| 39.9 |
> | LLaMA 3B    | GRPO          | 0.0 | 3.33 | 3.7   | 22.0|
> | LLaMA 3B    | Self Evolve   | 3.3 | 10.0| 5.3   | 29.3 |
> | LLaMA 3B    | E2H-G         | 3.3  | 10.0 | 5.3    | 28.6|
> | LLaMA 3B    | E2H-C         | 6.7  | 10.0 | 6.0    | 30.0 |
>
> We respectfully point out that LiveCodeBench (LCB) and agent-based tasks are beyond the scope of this work, which focuses on general and arithmetic reasoning. We leave their exploration to future work.
>
> > Q4. *How do the theoretical results apply to E2H vs standard curriculum techniques?*
>
> As clarified in our response to Q1 above, we do not perform task decomposition by breaking down a question into sub-questions. Instead, task decomposition operates at the dataset level, where we group questions by difficulty based on error rates. Hence, E2H is a general curriculum-based training method; our theoretical analysis under the Approximate Policy Iteration (API) framework applies broadly, whether the curriculum is implemented through probabilistic scheduling or fixed switching. Finally, our theoretical result holds for curriculum reinforcement learning in general, regardless of the specific scheduling strategy used.

---

> ### Author Response · Authors · 2025-11-21
> **Response to Reviewer kbdX 2/2**
>
> > Q5: Why do Qwen 1.5B and LLaMA 3.2 3B show different relative performance trends for E2H in Figure 1(a), especially at low k?
>
> Thank you for this insightful question. We confirm that all training artifacts and hyperparameters are identical across models. The difference in trends, particularly at low k, is due to architectural differences, such as the larger parameter count in LLaMA 3B compared to Qwen 1.5B. This hypothesis is supported by prior work, which shows that RL post-training leads to stronger performance gains for larger models at lower k, while smaller or base models tend to close the gap as k increases [4,5,6,7].
>
> References
>
> [1] LLM Reasoners: New Evaluation, Library, and Analysis of Step-by-Step Reasoning with Large Language Models, COLM 2024.
>
> [2] Inference-Time Computations for LLM Reasoning and Planning: A Benchmark and Insights, arxiv 2025.
>
> [3] OlympiadBench: A Challenging Benchmark for Promoting AGI with Olympiad-Level Bilingual Multimodal Scientific Problems, ACL 2024.
>
> [4] DeepSeekMath: Pushing the Limits of Mathematical Reasoning in Open Language Models, arxiv, 2024.
>
> [5] DeepSeek-R1: Incentivizing Reasoning Capability in LLMs via Reinforcement Learning, Nature, 2025.
>
> [6] SimpleRL-Zoo: Investigating and Taming Zero Reinforcement Learning for Open Base Models in the Wild, COLM, 2025.
>
> [7] Does Reinforcement Learning Really Incentivize Reasoning Capacity in LLMs Beyond the Base Model?, NeurIPS, 2025.

---

> ### Author Response · Authors · 2025-11-27
>
> Dear Reviewer kbdX,
>
> As the discussion phase comes to a close, we sincerely thank you for your time and thoughtful feedback. We have carefully considered your comments and provided detailed responses to each of your concerns in our earlier reply.
>
> We greatly appreciate your assessment of our work, and please don’t hesitate to let us know if there are any remaining questions or points we can clarify.
>
> Warm regards,
>
> Authors

---

### Meta-Review · Area_Chair_gPTJ · 2026-01-07

**Summary:**

This work proposes using a curriculum learning method where they schedule tasks from Easy to Hard during RLVR which shows better results at the end of training. The work combines task decomposition with probabilistic scheduling in CRL, addressing rollout inefficiencies in difficult reasoning tasks by building skills incrementally, which makes intuitive sense and extends prior RL post-training like DeepSeek-R1. The theoretical analysis provides finite-sample bounds and convergence guarantees, grounding the method in approximate policy iteration. The major concerns include lacking comparisons with DAPO, which seem to be well addressed in rebuttal.

**Reviewer Concerns:**

The major concerns include lacking comparisons with DAPO, which seem to be well addressed.

**Reviewer Scores:**

The average rating likely will increase.

---

### Decision · Program_Chairs · 2026-01-26

Accept (Poster)